# The First Domesticated ‘Cheongju Sorori Rice’ Excavated in Korea

**DOI:** 10.3390/plants13141948

**Published:** 2024-07-16

**Authors:** Yong-Gu Cho, Me-Sun Kim, Kwon Kyoo Kang, Joong Hyoun Chin, Ju-Kyung Yu, Soowon Cho, Chul-Won Lee, Jun Hyun Cho, Tae-Sik Park, Hak-Soo Suh, Mun-Hue Heu, Seung-Won Lee, Jong-Yoon Woo, Yung-Jo Lee

**Affiliations:** 1Department of Crop Science, College of Agriculture, Life & Environment Sciences, Chungbuk National University, Cheongju 28644, Republic of Korea; kimms0121@chungbuk.ac.kr (M.-S.K.); yujk0830@chungbuk.ac.kr (J.-K.Y.); cwlee@chungbuk.ac.kr (C.-W.L.); 2Institute of Agricultural Science & Technology, Chungbuk National University, Cheongju 28644, Republic of Korea; 3Division of Horticultural Biotechnology, Hankyong National University, Anseong 17579, Republic of Korea; kykang@hknu.ac.kr; 4Department of Integrative Biological Science and Industry, Sejong University, Seoul 05006, Republic of Korea; jhchin@sejong.ac.kr; 5Department of Plant Medicine, College of Agriculture, Life & Environment Sciences, Chungbuk National University, Cheongju 28644, Republic of Korea; soowon@chungbuk.ac.kr; 6National Institute of Crop Science, Rural Development Administration (RDA), Wanju 55365, Republic of Korea; hy4779@korea.kr (J.H.C.); jidaero.park@lge.com (T.-S.P.); 7College of Natural Resources, Yeungnam University, Kyongsan 38541, Republic of Korea; hssuh@ynu.ac.kr; 8Department of Crop Science, College of Agriculture and Life Sciences, Seoul National University, Seoul 08826, Republic of Korea; munhue@snu.ac.kr; 9Institute of Korean Prehistory, Cheongju 28763, Republic of Korea; arch15212@gmail.com

**Keywords:** early domesticated rice, Sorori rice, paleolithic, peat layer, quaternary geological layer, origin of rice

## Abstract

Archaeological excavations led by Yung-jo Lee and Jong-yoon Woo were carried out twice at the Sorori paleolithic site, Cheongju, in the Republic of Korea, at the upper stream of the Geumgang river, the Miho riverside. A total of 127 rice seeds were excavated, including 18 ancient rice and 109 Quasi-rice, in 1998 and 2001. At the first excavation, eleven short *japonica*-type ancient rice and one slender smooth ancient rice with two kinds of Quasi-rice were excavated. The average length of the 11 short rice grains obtained from the first and second excavation was 7.19 mm and the average width was 3.08 mm, respectively. The Quasi-rice are apparently different from the rice and do not have bi-peak protuberances on their glume surface. At the second excavation, six short ancient rice chaffs and some Quasi-rice 2 were found. These short-grained ancient rice were comparable to the ancient rice that were excavated at the Illsan Neolithic site. Geologists and radiologists confirmed that the peat layer in which the rice found was older than 15,000 years. In this study, the morphological characteristics, crushing, and DNA band patterns related to the genetic polymorphism of rice grains in Cheongju Sorori were compared and analyzed for genetic similarities and differences with wild rice, weed rice, and modern rice. The morphological, ecological, and physiological variations in rice grains excavated from the Sorori site were presumed to denote the origin of rice domestication in Korea. It is also suggested that the results of the DNA sequencing of excavated rice are very important clues in estimating the origin of the early domestication of rice.

## 1. Introduction

Plant domestication is a key evolutionary event where plant traits are genetically modified to select for characteristics that humans prefer and make plants more readily available for use. Domestication has had a major impact on food and nutritional security as it is accompanied by changes in important agronomic traits that affect plant productivity, adaptability, and quality. Research on the domestication process is important for guiding crop improvement efforts [1,2]. Rice, *Oryza sativa*, is one of the major grains in the world, especially in Asian food culture, and is used as a primary food source for half of the world’s population [3]. The rice species *Oryza sativa* is mainly the staple food in Asia, although the other species, *Oryza glaberrima*, has also been grown in Africa. These two species were independently domesticated as rice crops, *Oryza sativa* in Asia over 10,000 years ago and *Oryza glaberrima* in Africa over 3000 years ago. Today, Asian rice is the staple food for half the world, contributing an estimated 20% of human dietary calories [4]. The ancestor of Asian rice is the perennial wild relative of rice, *Oryza rufipogon*, and the ancestor of African rice is known as the perennial wild relative of rice *Oryza breviligulata*. It is estimated that these two species evolved through several stages from the same ancestor about 140 million years ago, and that geographic and reproductive isolation occurred around 2 to 3 million years ago, leading to speciation [4] (Figure 1). The origin of rice (*Oryza sativa*) cultivated in Asia is discussed in several regions according to the biological or archeological data. However, the most influential theory at present derives from Assam Yunnan Province as a result of investigating the habitat of the wild relatives of rice and intermediate ecotypes. Even now, many wild species are distributed in northern India, Nepal, Bangladesh, Myanmar, Vietnam, northern Thailand, and the Yunnan region in China, with the species being centered on this region [5,6,7,8].

Wild relatives of rice exist both as a perennial and as an annual plant, and both vegetative and seed propagation occur together. However, cultivated rice is an annual crop and reproduces via seed. Because the wild relatives of rice propagated by seed are cross-pollinated, wild relatives of rice have a wider range of environments for breeding than cultivated rice and have greater genetic variation. Characteristics that distinguish the wild relatives of rice or weedy rice include the angle of tillers, shape of seed, presence or absence of awns, length of anther, characteristics of the growth site and habit, lifespan of seed, dormancy, and shattering [9,10,11]. The lifeless excavated seeds are mainly determined using the shape of the seed (especially the length-to-width ratio), the presence or absence of awns, and the traces of shattering [10,12,13]. Seeds of wild relatives of rice have an abscission layer structure before they are completely ripe, so the seeds are easily dropped by light contact, such as weak wind, birds, or other animals. Because of these characteristics, wild relatives of rice have advantages in terms of survival, and usually the tip of the seed has a long awn about 10 times the size of the seed. The awn has elasticity, so it causes the dispersion of seeds and has the function of protecting seeds from birds or beasts. In addition, seeds of wild relatives of rice have a long dormant period and a wide germination period, so they can respond to unfavorable environments. As wild relatives of rice were domesticated to cultivated rice, the awn degenerated due to the “selective pressure of cultivation”, the dormancy was shortened or disappeared, and germination became uniform instead [14]. Asian rice is grown over a very wide area, from Mohe, Heilongjiang Riverside, China, at 53° north latitude, to the Rio Negro Riverside, Argentina, 40° south latitude, and it is also grown in highlands of up to 2600 m above sea level (Nepal, Jumila) [14]. As rice is known as a short-day plant, it is presumed that thermosensitivity and photosensitivity played a large role in its moving north from the subtropics known as the origin of rice to high latitudes. So far, rice hulls, carbonized rice, and traces of rice on earthenware have been excavated in Korea. The representative carbonized rice includes rice lumps (Three Kingdoms Period) from shell mounds in Gimhae [15], carbonized rice (BC. 850–410) excavated from a house site in Songguk-ri [16], carbonized rice (BC. 1270–860) excavated from a house site in Namgyeong, Pyongyang [16], and carbonized rice (BC. 1310–910) excavated from a house site in Heunam-ri, Yeoju [17]. Rice hulls known as rice seeds were excavated in Shinchang-dong, Gwangju; rice seeds were excavated in Gahyeon-ri, Gimpo [18]; rice seeds (4330 ± 80 BP.) were excavated in Gawaji District I, Goyang [19]; and rice seeds (2770 ± 60 BP.) were excavated in District II, in the same region.

The oldest carbonized rice was excavated in Sorori, Cheongju, Korea, in 1998. The Sorori paleolithic age site was noticed first by the team of Chungbuk National University Museum in 1998 through a survey of paleolithic tools buried in surface soil. Sorori is located at the low hill of Osong Mt, 2 km from the Miho river, which is the upper stream of the Geumgang river. The Sorori rice field is located in Cheongju, Chungbuk Province, in Korea (Figure 2). In Korea, 46 paleolithic sites have been found and they are either open or cave sites [12]. The rice field was located in a construction site for the Ochang Industrial Complex. During the period from 1997 to 1998, field surveys were undertaken to excavate cultural properties at this site. Many paleolithic stone tools, such as choppers, scrapers, points, and other artifacts, were also discovered. The investigation to search for ancient rice was begun in 1997 and completed in 1998. There were three excavation sites and three localities, A, B, and C, which were studied by the teams of Chungbuk National University Museum, Dankuk University Museum, and Seoul Municipal University Museum, respectively [20]. The Sorori site was located near the Geumgang river in Gongju and peat layers and 51 carbonized grains of rice samples were collected from this site.

The purpose of this study was to investigate the morphological and genetic characteristics of carbonized rice excavated from the first and the second excavations of the peat layer of the paleolithic site in Sorori, Cheongju. In addition, the agronomic, ecological, and genetic characteristics of rice seeds excavated in Sorori were examined, and the origin of Korean rice and the significance of the discovery of rice seeds in Sorori were considered.

Although there are multiple publications that address grain morphology, radiocarbon ages, PCR analysis, and genetic similarity, currently, there is no review that provides an overall report on the excavation, investigation, and analysis of the carbonized rice grains excavated in 1998 and 2001. This review aims to provide an overview of the rice excavation at Sorori, Cheongju, in the Republic of Korea.

**Figure 1 plants-13-01948-f001:**
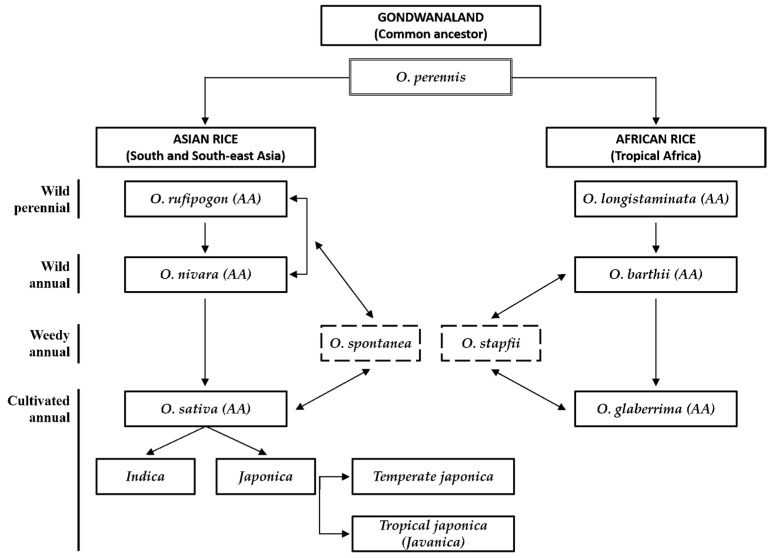
Schematic diagram of the evolutionary pathways of rice grown in Asia and Africa [21].

**Figure 2 plants-13-01948-f002:**
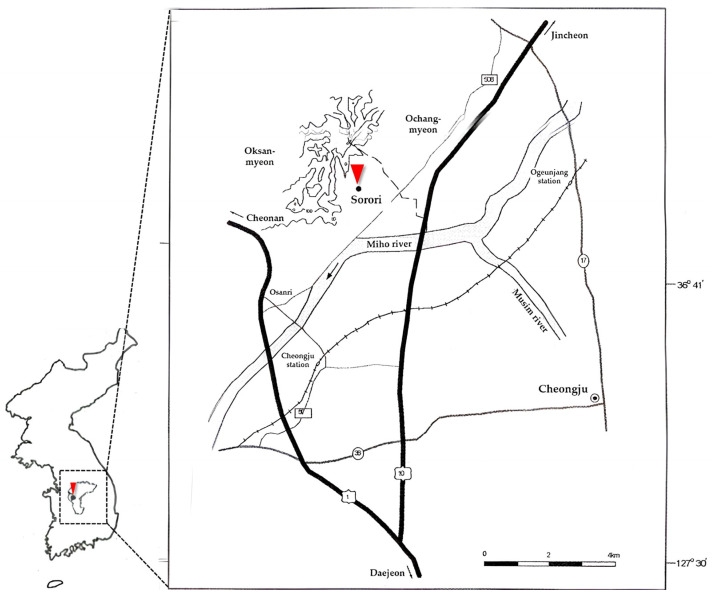
Location of Sorori site (▼) in Cheongju where the ancient rice was excavated.

## 2. Results

### 2.1. Archeological Excavation of Sorori Site

Archaeological excavations led by Yung-jo Lee and Jong-yoon Woo were carried out twice at the Sorori paleolithic site, Cheongju, in the Republic of Korea in 1998 and 2001. The Sorori paleolithic site is located at the Miho riverside, the upper stream of the Geumgang river. There were excavations of carbonized rice grains, some plants, and insects, along with ancient stone tools, during the archeological activities.

The action of excavating plant parts like grain, pollen, leaves, and stems, especially at an archeological site, is called the excavation of carbonized things. The methods of archeological excavation have to be extremely rigorous and only a few experts can complete this kind of work. A cross-sectional view of the area where rice seeds were excavated is shown in Figure 3. The area where the investigation was conducted was 26.1~34.6 m in peat area 2, and rice seeds were excavated in the 28.2~32.8 m section. By layer, Quasi-rice seed 1 (17,310 bp) is about 29.2 m, ancient short rice seed 1 (13,920 bp) is about 31.1 m, Quasi-rice seed 2 (14,820 bp) is about 31.8 m, and ancient short rice seed 2 (13,490 bp) is about 31.9 m. Quasi-rice 3 (12,780 bp) was excavated at about 32.0 m (Figure 3) [21].

At the first stage of the excavation work during November 1997–April 1998, many paleolithic tools were found in the cultural layer. Below the cultural layer, the peat layer was found under the clay profile where the soil wedges were inserted repeatedly.

Below the upper peat layer, there was about 1 m of deposited blue-gray clay. This was followed by a fine sand layer more than 1 m deep. In this layer, three thin, coarse sand layers were inserted; below this fine sand layer, the second peat layer was deposited, of about 70 cm in depth. Below the peat layer, there was the coarse sand layer above the riverbed [22]. At the second peat layer, 12 rice hulls and some grass seeds were found, and at the third peat layer, two different kinds of Quasi-rice were found in 1998 [22,23].

The geological analysis of soil and sediment profile was carried out by the team of the Korea Institute of Geology Mining and Materials. The profile of the soil and sediments deposited from 28 m to 37.5 m above sea level was diagnostically analyzed. The details are reported in the relevant papers. Radiocarbon age determination was carried out for the peat of both layers through the Geochron Laboratories, Chelmsford, MA, USA. The second peat layer showed 13,010 ± 190 bp and the third peat layer showed 17,310 ± 310 bp records [22]. Carbonized rice and insect fossils were excavated, such as Sorori ancient rice, Quasi-rice, and insect fossils, found in the Sorori peat layer (Figure 4).

### 2.2. Radiocarbon Measurements of Sorori Rice

A confirmation of the radiocarbon age of ancient rice was important for the establishment of a rice museum at the Sorori area. However, most remaining ancient remaining samples were treated with chemicals for conservation purposes.

Figure 5 shows the sampling location and the rice sample measured at the NSF Arizona AMS Laboratory and the stone artifact that might be used to scrape rice. There were only 18 ancient rice samples found from this site, but 13 samples were measurable for analytical purposes [24]. Six ancient rice samples that had been treated with preservatives were sent to the NSF Arizona AMS Laboratory in 2009. In addition, one invaluable rice sample which was embedded in a peat sample was sent to the laboratory for analysis. This rice grain had not been treated with any preservatives.

Sample pretreatment was performed using standard procedures of acid–alkali–acid (AAA) at the NSF Arizona AMS Laboratory. The dating was carried out for two separate batches of rice samples. Table 1 shows all radiocarbon ages obtained from the rices and peats investigated by the three AMS laboratories (Geochron, NSF Arizona Lab, and Seoul National University (SNU)) in 1998, 2001, and 2009 [25]. In Table 1, it can be seen that there are four layers which are associated with rice cultivation. These layers are rich in organisms and very dark [22,23].

From the new results, the first six samples were found to be modern and the second batch of rice and peat samples was old, dating to 12,500 bp. Since both rice and the surrounding peat had an identical age, 12,520 ± 150 bp for rice and 12,552 ± 90 bp for the peat layer, we were able to reconfirm that the radiocarbon age of Sorori rice is as old as 12,500 bp. The final results from the NSF Arizona AMS laboratory reconfirmed the radiocarbon age of Sorori rice [25].

### 2.3. Characteristics of Rice Hulls Found at First Excavation

Eleven short-grain-type and one slender-grain-type of ancient rice hull were found through floating some sandy soils at the first peat layer. This layer is located between 32.8 m and 30.8 m SL, and is as deep as 2.5 m beneath the surface soil. In the third peat layer, some clay and some small pieces of wood were found. While digging the clayish peat with a hand shovel, two kinds of grain hull were found, both looking like rice grains in appearance. After being washed, these two types of hulls showed clear differences. One was hairy and capsule-like and the other looked like a smooth, flat glume of grass panicles. These two kinds of hulls were named Quasi-rice 1 and Quasi-rice 2, respectively (Figure 6) [8,21,22].

#### 2.3.1. Ancient Rice Hull

Short-grain

There were eight whole grains (although partially damaged) and three split grains of the ancient short-type rice hull, which can be clearly identified as rice hulls by appearance. They are uneven in shape (see Table 2 measurements) and show different degrees of damage. As can be seen in (Figure 6A), they look morphologically different from the rice cultivated today: some are slightly similar to the rice excavated from Ilsan Family Area 1, and some are slightly similar to the current crop.

It is unknown whether this large variation in morphology comes from the difference in period or from the hybrid state of miscellaneous lines in the same era, and the variation is generally smaller than that between current cultivars. Observing the ridges on the young surface (Figure 7) with an electron microscope, all of them were found to have the characteristics of rice. The SEM picture clearly shows the bi-peak-protuberances on the surface of glume-like modern cultivars (Figure 7A) [8].

2.Long-grain rice

One long rice grain was excavated, and its size is similar to the current long-grain rice of IR 36, a modern *indica* cultivar (refer to the measured value), but its shape is unusual (Figure 6B) and it looks slightly different from ordinary rice. The gold hull is smooth and not hairy. The byeol is clear because there are no villi on the surface, making it genetically glabrous. It is possible to distinguish between the front and rear ridges, and there are pestles on the surface of the grain. It looks orange in the picture, but it is unclear whether it was a genetic gold hull, with this color in the living body. According to conventional wisdom, if long-grain rice is *indica* (although a DNA analysis could not be performed because there was only one sample), it is genetically recessive glabrous *indica* with a gold hull. To preserve its original shape, the DNA analysis was not undertaken, but an SEM picture of the glume surface clearly shows the same bi-peak protuberances of the rice hull (Figure 7B).

From the evolutionary point of view, this recessive gene was differentiated in ancient times, with the botanical and cultivation of rice, such as the differentiation of *indica* and *japonica*, and the differentiation of recessive mutations, which are now practical traits such as glabrous and gold hull. It is believed that this can provide an important indicator when pursuing the origin of rice types. It is a pity that a destructive analysis cannot be performed because there is only one sample [8].

#### 2.3.2. Quasi-Rices

3.Quasi-rice 1

A total of six half hulls were excavated, and their sizes vary widely. If two half hulls are put together, they may form a complete hull (Figure 6C). At first glance with the naked eye, the hulls look like rice, but when observed with magnification, the grains become like pods, the distinction between the outer and inner hulls of rice is unclear, and the vertical surface is not visible. However, the hulls look similar to the structure of rice grains because of the clear traces of guards and secondary guards. When was engrafted in the clay–peat, they looked like rice hulls, but after being washed and dried, they turned out to be a capsule instead of glume. Their size was variable; some were smaller or larger than the currently recommended *japonica* cultivars. The SEM picture does not show the bi-peak protuberances (Figure 7C).

Following observation with an electron microscope, the structure of the rice hull was not visible at all (Figure 7), so it could not be concluded that it was rice; therefore, it will be referred to as “Quasi-rice 1” here. Oeyoung’s pestle can distinguish all wild species, as well as cultivated rice, so it seems to be a good indicator for distinguishing rice genuses at present [8].

4.Quasi-rice 2

Two complete, five incomplete, and many broken pieces of glume-like hulls were found in the clay–peat. These were also engrafted in the clay–peat. They looked like rice hulls when they were in the soil in a complete pair. However, after being washed and dried, they looked entirely different. Many of them were in broken pieces and only two were in complete pairs. Their size was comparable to the currently recommended cultivars. The SEM picture does not show the bi-peak protuberances and a DNA analysis showed quite a different picture (Figure 7) [8].

#### 2.3.3. Morphological Variations in Grain Size and Length/Width Ratio

The two groups of rice and two groups of Quasi-rice showed a peculiar morphology although there were variations in size. Both short and long rice grains seemed to have a very different shape in comparison to present-day cultivars. The variation in the short-grain type within a limited area, like in this excavation, might imply a primitive evolutionary stage of plant fauna. Although we could not see the variations, due to there being only a single sample of slender-grain rice, the morphology of long-grain rice was distinguished from the morphology of short-grain rice and also from the morphology of long-grain rice from present-day cultivars. Quasi-rice 1 and Quasi-rice 2 looked different from the rice hulls existing currently. Naturally, the evolutionary relationship between these Quasi-rice and the current rice species, including wild species and cultivars, are not understood yet (Figure 7) [8].

Although the number is small, 13 intact (measurable) rice seeds excavated from Sorori were compared in terms of size and grain shape with rice seeds excavated from Gawaji in Korea of about 3000 BP (District II) and 5020 BP (District I).

We compared the 13 Sorori ancient rice grains with Gawaji rice seeds (Type I 5020 BP, Type II 3000 BP) and created a distribution map, as shown in Figure 8. The rice seeds excavated from Sorori showed large morphological variation, were slightly larger than the rice seeds excavated from Gawaji, and were closer to *Japonica*, with one grain being close to *Indica* and one grain estimated to be similar to tropical *Japonica* (*javanica*, Appendix A). Although there is a long gap between the two excavated rice seeds, if we assume from these results that the Sorori-excavated rice seeds are genetically connected to the Gawaji-excavated rice seeds, it can be assumed that the side that was genetically mixed became uniform with the slightly smaller and slightly longer side. It was presumed that human selection pressure or the pressure to obtain sufficient survival to cultivate rice were involved in this process, but this could only be inferred through more relics from various places [24].

#### 2.3.4. Bi-Peak Protuberances on the Surface of Hull

The SEM picture shows bi-peak protuberances on the surface of the hull of six rice species including *O. sativa*, *O. glaberrima*, *O. rufipogon*, *O. barthii*, *O. brachiantha* and *O. nivara* (Figure 9). And a grass species, *Leersia japonica,* was observed to have contrasting results. Needless to say, both *japonicas* and *indicas*, all the *Oryza* species observed so far, showed the bi-peak protuberances, while the *Leersia japonica* was flat with some shallow ridges in the crossways. The SEM picture of excavated rice samples, both short and long, showed bi-peak-protuberances, while both Quasi-rices did not show the protuberances. Instead, they showed flat shallow ridges similar to that of *Leersia japonica* [8].

#### 2.3.5. DNA Variations in the Carbonized Rice

In order to compare the grouping with the current rice by analyzing the DNA of six types of carbonized rice seed excavated from the peat layer of the Sorori site in Cheongju, URPs (universal rice primers) were used. The URPs were derived from the rice’s repetitive DNA sequence of pKRD (GenBank accession No. AF241234). The primers, with a length of 18~20 nucleotides, were designed using primer designer V.1.01 (Scientific and educational software) from both strands of the sequences consisting of nucleotides of 1187 bp [26].

For PCR analysis, two ancient short-grains, four Quasi-rice seeds, *Indica* rice IR36, Peh-kuh, *Japonica* rice Nakdong, Taichung 65, single-grained weedy rice Gyeongsan Aengmi 2, Suncheon Aengmi 1, short-grain weedy rice Hapcheon Aengmi 3, Seongju Aengmi 8, wild relatives of rice W1944 and W130, and carbonized rice seeds excavated from Gawaji, Ilsan, were used, as shown in Figure 10. DNA extracted from the seed chaffs was subjected to PCR analysis using seven primers, including URP1, URP2, URP4, URP6, URP12, URP13, and URP15.

As a result of the PCR amplification of DNA extracted from seed chaffs using seven universal primers, various band patterns were shown when URP1, URP12, and URP13 were used, as shown in Figure 11. However, when URP2, URP4, URP6, and URP15 were used, almost identical band patterns were shown, and the DNA region that was homologous with these primers was revealed to have DNA bands that exist in both modern rice and ancient rice seeds.

We performed PCR reactions using the primer sets URP1, URP12, and URP13 on DNA from 17 samples, which included ancient rice seeds and modern rice tissues. The PCR results, based on the discerned base pair sizes of the amplified PCR bands obtained using URP1, URP12, and URP13, were analyzed using the UPGMA method implemented in POPTREE2 software [27] to assess the genetic similarities between ancient and modern rice varieties.

The 17 samples were grouped into four clusters. The first cluster included Quasi-rice 1, Quasi-rice 2, Quasi-rice 3, ancient short-grain, and IR 36. The second cluster comprised Peh-kuh, W1944, Hapcheon Aengmi 3, and Suncheon Aengmi 1. The third cluster contained Taichung 65, Seongju Aengmi 8, Quasi-rice 4, Gyeongsan Aengmi 2, and W130. The final cluster grouped Nakdong (a *Japonica* rice variety) with Gawaji excavated rice [28] (Figure 12).

#### 2.3.6. Evidence That the Sorori Environment Was a Wetland through the Identification of Beetle Fossils from the Sorori Peat Bed

The fact that the ancient rice seeds from the Sorori peat layer are approximately 15,000 BP years old was revealed through various direct and indirect evidence, including radiocarbon dating. However, foreign experts are questioning whether the environment at that time was suitable for rice growth: was the minimum temperature suitable for growth even at the end of the Ice Age, and was a wetland environment suitable for rice growth actually created [29].

During an investigation of ancient rice seeds in the peat layer of Sorori site, fragments that appeared to be fragments of insects were excavated (Figure 13A) along with various other organic materials. Although it was not found as a complete object, the exoskeleton of insects made of chitin is quite stable; the exoskeleton of beetles is particularly thick and hard, so it seemed to have been preserved. In addition, these exoskeletons were characteristics of beetle species, so some identification was possible. As a result of a comparative analysis with modern species based on the shape of the forewing, the species was identified to be the same as or very closely related to the two modern species *Donacia flemola* and *Plateumaris sericea* (Figure 13B, Figure 14 and Figure 15). These are insects belonging to Coleoptera, Chrysomelidae, and Donacinae, and were found to mainly attack wetland plants such as Cyperaceae and Poaceae.

Based on these insects’ detailed ecological information, it is clearly indicated that the Sorori environment was a wetland at the time. Therefore, it is clear from the evidence of insect fossils as well as ancient rice seeds that the environment of Sorori at the time was a favorable environment for growing wetland plants such as rice [29].

### 2.4. Characteristics of Ancient Rice Grains Found at Second Sorori Excavation

To clarify the existence of ancient rice grain remains at the Sorori site, a second excavation was carried out in 2001. In the excavation, geologists confirmed there were three layers of peat soil. In the second peat layer, six rice grains and some Quasi-rice were found. A few Quasi-rice were found in the third peat layer too. Among those, only four rice hulls and 14 Quasi-rice whose complete shape was maintained were observed [8].

The rice hulls looked the same as those of the “ancient short-grain rice” and the Quasi-rice was the same as the Quasi-rice found at the first excavation. The “ancient slender-grain rice” and “Quasi-rice 1” were found at the first excavation, but were not found at the second excavation. The rice hulls and Quasi-rice 2 found at the second excavation looked similar to the ones found at the first excavation, and the rice hull looked similar to those of current cultivars in size and shape (Table 3 and Figure 16). The rice hull looked similar to those of some wild forms of *Oryza sativa* in the size and shape of the glume. The DNA analysis carried out by Dr. H.S. Suh showed that the “ancient short-grain rice” showed approximately 30% similarity with current *japonica* cultivars [30]. Based on these facts, we would like to wait to judge the species of this rice hull. At the peat layer where the rice hull was found, no accompanying materials such as a rice straw or tools for rice cultivation or utilization were found. Based on these facts, we also would like to wait to judge whether this rice is cultivated or not.

The carbon dating carried out at the Seoul National University (SNU) Radiology Laboratory showed 12,930 ± 400 bp and 13,920 ± 200 bp for the samples taken from the upper portion and lower portion of the second peat layer, respectively [31].

### 2.5. Comparison of Ancient Rice Grains Found at Ilsan Site I

The Ilsan archeological site was excavated by Prof. Sohn bo-gi in 1991. It is located at Daehwa 4-Dong, Ilsan city, Kyunggi Province (N 37°41′/E 126°44′) [32].

Besides site I, there were two more sites excavated at the same time. Each of the carbonized rice types was also found in these sites. Ten carbonized rice grains were excavated at Ilsan site I. Among these, two grains were destroyed and the rest had a complete shape, and their length and width were measured (Table 4) [32]. The appearance of the grains was not exactly like current *Japonica* cultivars. The report of the excavation explained the shape of the grains in detail and named the grains “ancient rice”. The explanation of the grain shape says that the grain appears generally longer than current *japonica* cultivars and slender near the pedicel side and broader near the apex side. At the tip, many bristle hairs and distinct awn scars are present. A few of them seemed to have a smooth hull. Some of them were threshed at the rachilla instead of at the pedicel and most of them showed longer rachilla with longer outer glumes (Figure 17). Based on these facts, it was presumed that the morphology of the rice was less uniform, and the awnless and smooth hulls were already differentiated by that period [8].

### 2.6. Significance of Sorori Ancient Rice Grains

The size of the ancient rice grains excavated at Sorori site and Ilsan site are comparable with those of current cultivars. No awned grains were found at the Sorori site, unlike at the Ilsan site, where one awned rice grain was found. The general shape of the excavated rice looks different; for example, grains narrower at the shoulder and broader at the middle, and some bristle hairs (stronger than current cultivars) are present at the tip of the grain. These grains were named “ancient rice” in the report of the Ilsan rice excavation. Sorori rice showed much wider variations in the size and shape of grains and, notably, one smooth, slender gold hull. Almost the same grain type was reported at the Danyang “Suyanggae” site [33]. This raises many questions about the evolution of rice farming.

At the second excavation, slender rice and Quasi-rice 1 were not found. But several ancient rice and many Quasi-rice 2 grains were found at the same location. This might mean that if someone digs again around the same place, they might harvest some more ancient rice with of the Quasi-rice 2 type. The age survey with C-14 conducted at a different institute showed considerable agreement with Quasi-rice 2 and peat soil, but the survey with the rice itself has not been conducted yet. Some weed seeds and insect remains were also found in the same layer of peat where the rice was found. However, none of the tools for rice-growing and no rice plant remains were found in the same peat layer (Table 5) [31]. Removing the husk is essential for the utilization of rice. We wish to see the tools that might have been used on the rice grains so that the debate regarding whether the rice was grown locally or imported would be settled.

### 2.7. Agronomic Significance of Ancient Rice Seeds Excavated from Sorori

The stages in which humans cultivate crops from wild plants are divided into domestication, cultivation, and agriculture. These processes are applied to rice and can be used to determine the “origins of cultivated rice” and “rice farming”; this is referred to as “rice farming culture.” There is a view that these three stages are qualitatively different and that it is not easy to determine the acclimation process [34], but one version divides the cultivation of wild plants into three stages: (1) gathering and hunting stage, (2) semi-cultivation stage, (3) early farming (domestication of cultivated plants) stage, and (4) establishment of agriculture (differentiation of cultivated plants) to establish a complete symbiotic relationship between humans and plants [24].

When looking at Sorori ancient rice seeds in light of these stages of development, I would like to consider what stage has been preserved, focusing on the arguments discussed earlier. Considering Sorori ancient rice seeds, based on the research results so far, the natural environment surrounding the excavation site (Peat II area, Sorori A district) is a swampy area capable of cultivating rice, and the frequency of excavation is significant (six grains were collected from 6.5 m^2^ during the second excavation) (Figure 18).

The ancient rice seeds have the characteristics of cultivated rice, with no awn and less seed shattering, and have something in common with the later-cultivated rice, the rice excavated from Gawaji. However, the rice seeds are genetically very diverse and have a large range of variation. The Sorori region has maintained a climatic environment in which rice can grow and reproduce well since the time when Sorori ancient rice seeds were grown; however, currently, wild relatives of rice (*O. rufipogon*, *O. nivara*, etc.), which can be considered the ancestor of rice, do not exist in Korea, and only weedy rice exists. Considering that Sorori ancient rice seeds are the oldest (15,000 BP) among the extant rice relics, they are presumed to represent a stage of natural selection during the origin of rice, the process of humans bringing seeds to the field, sowing, and harvesting them. In other words, it is presumed that they were subject to strong pressure from the humans who lived on the Korean Peninsula [24].

As a result of investigating ancient rice seeds excavated from Sorori with SEM, it was possible to observe two cases where the rice plants had not fallen off (Figure 19A) and one or two cases where the abscission layer of the attached rice grain was rough. The fact that the pedicel remains means that rice was harvested by humans when it was quite ripe. Early farmers took rice grains from the panicles when harvesting (Figure 19B). Grains had to be removed from the rice panicles by hand or using some primitive tool, and in the process, it is thought that when the grains were forcibly removed from the panicle, the pedicel of the grains became tattered and broken, as shown in Figure 19C. The appearance of the pedicels, which look like the rice grains were forcibly cut off, is because rice was cultivated at the time and farmers used their hands or tools to harvest the rice grains during the harvest season, irregularly tearing the pedicels.

If immature rice grains had been harvested, the silicification of the rice seeds (chaff) would have not been completed, so it would have decomposed within a few years and would have not been preserved for 15,000 BP. This is considered to be evidence that the rice seeds excavated from Sorori clearly have the characteristics of cultivated rice, and a more certain judgment could be made by examining all the excavated rice seeds [24].

Questions have been raised about whether a wetland environment suitable for growing rice was actually created. However, fossil insects such as beetles have been discovered along with ancient rice seeds (Figure 13), and this evidence has made it clear that the environment at that time was a wetland with the environment and food requirements for these root leaf beetles to grow and reproduce. This fact shows that the environment in which the ancient rice seeds were excavated was a wetland environment conducive to growing rice, and thus is also evidence of its potential as the origin of an ancient agricultural culture.

Therefore, citing this theory, the authors would like to interpret the grains as rice that was in the process of domestication, between the semi-cultivation stage and the early agricultural stage. In other words, Sorori ancient rice is an ancestor of Korean cultivated rice, and can be considered to be rice in the early stage of domestication. Also, if we follow the theory that cultivated rice originated in several places, as advocated by Yu Mun-myeong [35], it seems that Sorori can also be recorded as an origin of cultivated rice.

## 3. Discussion

### 3.1. Excavation of ‘Cheongju Sorori Rice’

The Sorori Paleolithic-age site was excavated in the course of a site survey for cultural properties at the pre-arranged area for the Ochang Industrial Complex by the team of Chungbuk National University Museum in 1997~1998. Sorori (a small village in Cheongju city, Chungcheongbukdo, Republic of Korea) is located at the low hill of Osong Mt., some 2 km apart from the Miho river, which is the upper stream of the Gumgang river. During the excavation, Paleolithic artifacts like a cleaver, scraper, notch, core, and flakes were gathered. Moreover, the quaternary geological layers were observed to be well-preserved. Through a preliminary test excavation to ascertain the presence of a cultural layer, not only Paleolithic layers but thick peat layers were identified. So, the team carried out the first excavation of the Paleolithic and peat layers during January 1997~April 1998. The second excavation was carried out for the first peat layers in the Sorori A-II area for 2 months from September 2001 [24].

The Sorori A-II peat area is formed of three peat layers. Samples of the first (two dating sample, 9500 bp), second (14 dating sample, 12,500~14,800 bp), and third peat layer (five dating samples, 16,300~17,300 bp) was analyzed at the Geochron Lab (USA), University of Arizona Lab (Tucson, AZ, USA), and Seoul National University Lab (Seoul, Republic of Korea). Before long, we obtained the “SAME ABSOLUTE DATINGS” from these three AMS labs [25]; consequently, it is clear that the deposit of peat layers is “SO STABLE” (Table 1, Figure 5). In the first investigation, ancient rice seeds were found in the second peat layer and Quasi-rice in the third peat layer. Additionally, we confirmed that carbonized ancient rice and carbonized Quasi-rice were concentrated in a certain spot (in situ) through the second investigation, which means the materials had not been contaminated. One of the Quasi-rice in the upper part of the second peat layer was directly dated at 12,500 bp. This shows that the dating of the rice itself corresponds to that of the peat layer. Additionally, Carabidae was found among many floral and insect data, which is known to live on the stalk of the family Gramineae during the larval stage; this insect was interpreted to be related to the presence of rice. Eight complete grains and three broken grains of ancient short-type rice hulls were found in the first excavation. Their shape and size showed variation and the severity of weathering was also variable (Figure 6 and Figure 7).

### 3.2. Morphological Variations in Carbonized Ric

Most of the complete grains showed a somewhat similar morphology to Gawaji ancient rice. Weathering was severe and no remains were observed inside the hull. The SEM picture clearly showed the bi-peak protuberances on the surface of the glume, like modern cultivars (Figure 18) [8]. One slender-grain hull was found. The shape and the size were slightly different from those of IR 36, a modern *indica* cultivar. The gold hull was smooth and not hairy. The SEM picture of the glume surface clearly showed the same bi-peak-protuberances in the rice hull (Figure 9).

At the second peat layer, four complete and two broken rice seeds and some Quasi-rice grains were found in the second excavation. The average length of the 12 short-grain rice was 7.19 mm and the average width was 3.08 mm (Table 2). According to Lu et al. [36], when the grain length and width of 127 rice varieties were investigated, they presented with substantial variations in grain size, with grain length ranging from 6.32 to 12.95 mm and grain width ranging from 2.29 to 4.43 mm. The 12 Sorori seeds had close to the shortest grain length (6.32 mm), at 7.19 mm, and the grain width was also close to the smallest (2.29 mm), at 3.08 mm.

Loss of the seed-shattering habit is thought to be one of the most important traits in rice cultivation, as the “easy-to-shatter” nature of wild rice severely reduces yield during the harvest season. Throughout human history, distinct grain threshing systems have developed in different eras in accordance with the degree of seed-shattering. In current rice breeding programs, this seed-shattering habit remains a target, especially in the establishment of new *indica* (another subspecies of *O. sativa*) varieties. Therefore, the seed-shattering habit is one of the most important agronomic traits in rice cultivation and breeding [37]. Comparing Sorori ancient rice grains with Gawaji rice excavated at the lower reaches of Han River in Koyang City by Dr. Yung-Jo Lee, Chungbuk National University, in 1999, the Sorori rice grains showed more variation in the distribution of size and shape than the Gawaji rice grains (Figure 17). Also, the Sorori ancient rice grains exhibited awnless, rough abscission scars and a rudimentary glume and pedicel, which suggests less shattering [22]. The tattered and broken pedicel of the excavated ancient rice might mean that the rice was harvested by man when it was almost ripe (Figure 19C). Early farmers trashed grains from the rice panicles when harvesting (Figure 19B). Grains had to be removed from the rice panicles by hand or using some primitive tool; during this process, it is thought that when the grains were forcibly removed from the panicle, the pedicel of grains appeared tattered and broken, as shown in Figure 19C. The appearance of the pedicels, which look like the rice grains were forcibly cut off, is because rice was cultivated at the time and farmers used their hands or tools to harvest the rice grains during the harvest season, which presumably irregularly tore the rice pedicels [24,38].

### 3.3. DNA Variations in Carbonized Rice

The DNA from six carbonized rice seeds excavated from the peat layer of the Sorori ruins in Cheongju was analyzed using PCR to compare their genetic similarity with modern and weedy rice varieties, utilizing universal rice primers (URP). Various band patterns were detected with the URPs URP1, URP12, and URP13, as shown in Figure 11. Based on the base pair sizes of the amplified PCR bands obtained using these primers, clustering between ancient rice seeds, wild relatives, weedy rice, and modern rice was analyzed using the UPGMA method using POPTREE2 software [27] (Figure 12).

One ancient short-grain rice sample (15) grouped with Quasi 1, Quasi 2, and Quasi 3, and was closely related to the currently cultivated *indica* rice, IR 36. Another ancient short-grain rice sample (12) also clustered near Quasi 1, Quasi 2, Quasi 3, and IR 36. Quasi-rice 4 was clustered near the weedy rice Seongju Aengmi 8 and the *japonica* variety Taichung. Carbonized rice seed excavated from Gawaji was clustered with Nakdong, a *japonica*-type rice cultivated in Korea.

Previous reports on DNA analysis based on the sequencing of carbonized rice grains used DNA from 2500–1500-year-old grains [39]. However, this study utilized only the empty hulls of 15,000-year-old rice samples, which contained very little DNA. Two ancient short-grain rice samples (12 and 15) and three Quasi-rice samples were clustered together with the *indica* rice IR 36. However, Quasi-rice 4 clustered with *japonica* rice Taichung 65 and weedy rice Seongju Aengmi 8.

The DNA analysis did not clearly determine the ecotype of Sorori rice. It is assumed that ancient rice populations began as mixed populations and gradually evolved into ecotypes of *indica* or *japonica*, with an increasing proportion of individual plants adapting to environmental changes. It is also hypothesized that Sorori rice represents ‘a differentiating type of rice’ in the process of divergence into weedy rice, *indica*, and *japonica* varieties. This suggests that Sorori rice could be considered ‘the origin of native Korean rice’ [30,38].

### 3.4. Agronomic Importance of Ancient Rice

Although questions might be raised about whether a wetland environment suitable for growing rice was available following the excavation of the carbonized rice, fossil insects such as beetles have been discovered along with ancient rice seeds, and this evidence has made it clear that the environment at that time was a wetland with the environmental and food requirements for these root leaf beetles to grow and reproduce. This fact shows that the environment in which the ancient rice seeds were excavated was a wetland environment conducive to growing rice, and thus is also evidence of the environment’s potential as the origin site of an ancient agricultural culture.

The ancient rice seeds have the characteristics of cultivated rice, with no awn and less seed-shattering and have something in common with the later cultivated rice (5020 BP), the rice excavated from Gawaji. However, rice is genetically very diverse and has a large range of variation. The Sorori region has maintained a climatic environment in which rice can grow and reproduce well since the time when Sorori ancient rice seeds were grown. Considering that the Sorori ancient rice seeds are the oldest (15,000 BP) among the extant rice relics, they are presumed to represent a stage of natural selection during the process of cultivation, where humans brought seeds from the origin of rice, sowed, and harvested them [24].

When ancient rice seeds excavated from Sorori were investigated via SEM, it was possible to observe two cases where the rice plants had not fallen off (Figure 19A) and one or two cases which the abscission layer where the end part of the rice grain was attached was rough. The fact that the pedicel remains means that the rice was harvested by man when it was quite ripe. Early farmers trashed grains from the rice panicles when harvesting (Figure 19B). Grains had to be removed from the rice panicles by hand or using some primitive tool; during this process, it is thought that when the grains were forcibly removed from the panicle, the pedicel of grains appeared tattered and broken, as shown in Figure 19C. This is considered to be evidence that the rice seeds excavated from Sorori clearly had the characteristics of cultivated rice, and a more certain judgment could be made by examining all the excavated rice seeds [24]. Therefore, the author would like to interpret the findings as rice that was in the process of domestication, between the semi-cultivation stage and the early agricultural stage. In other words, Sorori ancient rice is the ancestor of Korean cultivated rice, and can be interpreted as rice in the early stage of domestication.

The natural and long-documented presence of weedy rice provides further proof that the wild-type rice are descendants of the ancient rice found in the Korean peninsula. This weedy rice can now be discovered anywhere in the Korean peninsula lowland, showing the grain shape of both the *japonica* and *indica* types and the genetically mixed characteristic of both ecotypes [23]. The variety name ‘Sororido’, the same name as Oksan Sorori in Cheongju, where the ancient rice remains were excavated, was last noted in the report listing Korean local rice varieties collected in 1911~1912 [21]. The locations where the local rice variety ‘Oksando’ was collected included several areas, like Eumseong in Chungcheongbukdo, and Icheon, Pocheon, Juksan, Yong-in, Yeoju, Gwangju, Suwon, Yangju, Pyeongtaek in Kyeonggido, and Byeokdong in Pyeong-an-bukdo of North Korea. These local rice varieties collected in 1911~1912 might be further evidence that such rice originated from the Sorori ancient rice.

Also, if we follow the theory that cultivated rice originated in several places, as advocated by Yu Mun-myeong [35], it seems that Sorori could also be recorded as an origin of cultivated rice. The terminology of ‘first domesticated rice’ was described in the ‘5th and 7th *Book of Archaeology*’ in 2005 and 2017, as shown in Appendix A. In conclusion, according to the morphological, ecological and genetical variations in the excavated rice grains from the Sorori site, the origin of early domesticated rice in Korea can be estimated. It is suggested that the estimation of the origin of early domesticated rice should be further confirmed by a comparative analysis based on DNA sequencing of the excavated rice and the current rice [40].

## 4. Materials and Methods

### 4.1. Analysis of Morphological Characteristics Using Carbonized Rice Hull

The length, width, and thickness of excavated rice seeds were investigated using Mitutoyo^®^ Vernier Calipers (CD-20CP, Kawasaki, Tokyo, Japan), and the length-to-width ratio was calculated as the ratio of length to width. Seeds with a length of 7.5 mm were classified as extra-long-grain, 6.6–7.5 mm as long-grain, 5.51–6.6 mm as medium-grain, and 5.5 mm or less as short-grain [40]. The length-to-width ratio led grains to be classified into a round type when 1.0 or less, a bold type when 1.1–2.0, a medium type when 2.1–3.0, and a slender type when 3.0 or more [41].

### 4.2. Histological Analysis

A scanning electron microscope (SEM) (SUPRA 55VP, Zeiss, Jena, Germany) was used for histological analysis of the excavated rice seeds, and the seed structure was observed by enlarging the middle part of the rice hull. The sample was fixed with 2.5% glutaraldehyde solution for 3 h, and then washed three times with 0.1 M phosphate buffer for 20 min each. Samples were dehydrated with a graded series of ethanol (30, 50, 70, 80, 90, 95, 100%) for 15 min each, and the dehydrated samples were substituted with 100% isoamyl acetate for 20 min. After drying the sample in a clean bench, it was coated with platinum and observed under a scanning electron microscope [42].

### 4.3. Extraction of DNA from Carbonized Rice Hull

DNA was extracted from four pieces of carbonized rice hull excavated from the peat soil layer aged 13,010 ± 190 bp, and two pieces of carbonized Quasi-rice hull excavated from the layer aged 17,310 ± 310 bp. Half of the carbonized rice was homogenized with 750 µL of TEN 8 (100 mmol EDTA, 500 mmol NaCl, pH 8.0) and the homogenate was centrifuged at 12,000 rpm for 10 min via micro-refrigerated centrifuge. The supernatant was mixed 30 µL of 20% SDS and heated at 65 °C for 20 min. After centrifugation at 12,000 rpm for 5 min, 500 µL of a supernatant was mixed with an equal volume of cold iso-propanol and DNA was further purified via Sephadex G-50 spin column (Thermo Fisher, Waltham, MA, USA) [43,44]. The DNA pellet recovered by ethanol precipitation was dissolved in 10 µL of distilled water before use.

* bp: radiocarbon age, BP: recalibrated age.

### 4.4. Amplification of Ancient DNA Fragment by PCR

A total volume of 25 µL reaction mixture was completed with 2.5 µL of 10× PCR buffer, 0.2 µL primer, 0.2 mmol dNTPs, 1 unit of Taq DNA polymerase (Promega, Madison, WI, USA), and 2 µL of ancient DNA from the carbonized rice and/or carbonized Quasi-rice hull. Seven Universal Rice Primers (URPs, https://www.seoulin.co.kr/) [26] were used for DNA amplification (Appendix A). The reaction mixture was subjected to Perkin Elmer 2400 thermal cycler (GeneAmp 2400, Waltham, MA, USA) for 45 cycles consisting of 1 min at 94 °C, 1 min at 37 °C, and 2 min at 72 °C, followed by an extension of 5 min at 72 °C. A total of 5 µL of the 1st PCR product was subjected to the 2nd PCR with the same process. Amplified DNA fragments were analyzed by 1.4% agarose gel electrophoresis followed by EtBr staining.

Two ancient short-grains, four Quasi-rice seeds, *Indica* cultivated rice IR 36, Peh-kuh, *Japonica* cultivated rice Nakdong rice, Taichung 65, short-grained weedy rice Gyeongsan Angmi 2, Suncheon Angmi 1, short-grained weedy rice Hapcheon Angmi 3, Seongju Angmi 8, wild relatives of rice W1944 and W130, and carbonized rice seeds excavated from Gawaji, Ilsan, were analyzed (Figure 14). The analysis of DNA primers was designed based on rice-derived repetitive DNA sequences synthesized with 18 to 20 bp (base pairs) lengths. Seven primers, including URP1, URP2, URP4, URP6, URP12, URP13, and URP15, are universal primers (https://www.seoulin.co.kr/) [26]. DNA extracted from the seed shell was amplified by PCR (polymerase chain reaction), and their mutual relationship was analyzed.

### 4.5. Genetic Similarity Analysis

For genetic similarity analysis, two ancient short-grains, four Quasi-rice seeds, *Indica* rice IR36, Peh-kuh, *Japonica* rice Nakdong, Taichung 65, single-grained weedy rice Gyeongsan Aengmi 2, Suncheon Aengmi 1, short-grain weedy rice Hapcheon Aengmi 3, Seongju Aengmi 8, wild relatives of rice W1944 and W130, and carbonized rice seeds excavated from Gawaji, Ilsan, were compared to determine the variations in three universal rice primers (URP). Genetic similarity was analyzed using the UPGMA method of POPTREE2 software [27]. The allele sizes were scored in base pairs (bp) based on the relative migration of the internal size standard. The determination of allele sizes was performed using Microsoft Excel 2021 software. A phylogenetic analysis of ancient rice *japonica* varieties, and *indica* varieties was performed with genotyping data from three URPs using the unweighted pair group method with arithmetic mean (UPGMA) dendrogram (1000 bootstraps), using POPTREE2 software. Genetic distance was calculated using “Nei’s standard” [45,46].

### 4.6. Identification of Coleopteran Insect Fossils Found in Sorori Peat Layer

At the beginning of the research, peat lumps were dissolved in water to filter out insect fossil fragments [47] (Figure 18; however, in the process of excavating fossil insects after 2001, unlike before, the peat lumps were carefully broken apart to remove the insect fossils that appeared on the surface. By storing them as-is, without dissolving them in water [29], and using them as data, broken and weakened insect fossil fragments were prevented from breaking or scattering again (Figure 13A). Later, the insect fragments investigated in this way went through a detailed process of being re-dissolved in water and filtered for a more detailed investigation. The fossils obtained through this excavation were difficult to identify because, in most cases, one wing or part of the body was crushed and broken, rather than the entire insect. The investigated fossils were first divided into groups of insects that were judged to be similar or identical, and in cases where comparative analysis was difficult because they were partial fossils, they were classified separately. Taxonomically, we first investigated to which insect order it belonged within the insect class, and then, for available fossils, we investigated the family, and further, the genus and species. The standard for the species identification of insects is based on current species. In particular, among beetles, species identification may be possible through a micromorphological analysis of forewings that are in relatively good condition. In this survey, intensive research was conducted focusing on identification in particular. This was accomplished through direct discussion with experts in the field, including Dr. Ahn Seung-Lak, director of the Natural History Research Department at the National Science Museum, and specimens were confirmed.

## Figures and Tables

**Figure 3 plants-13-01948-f003:**
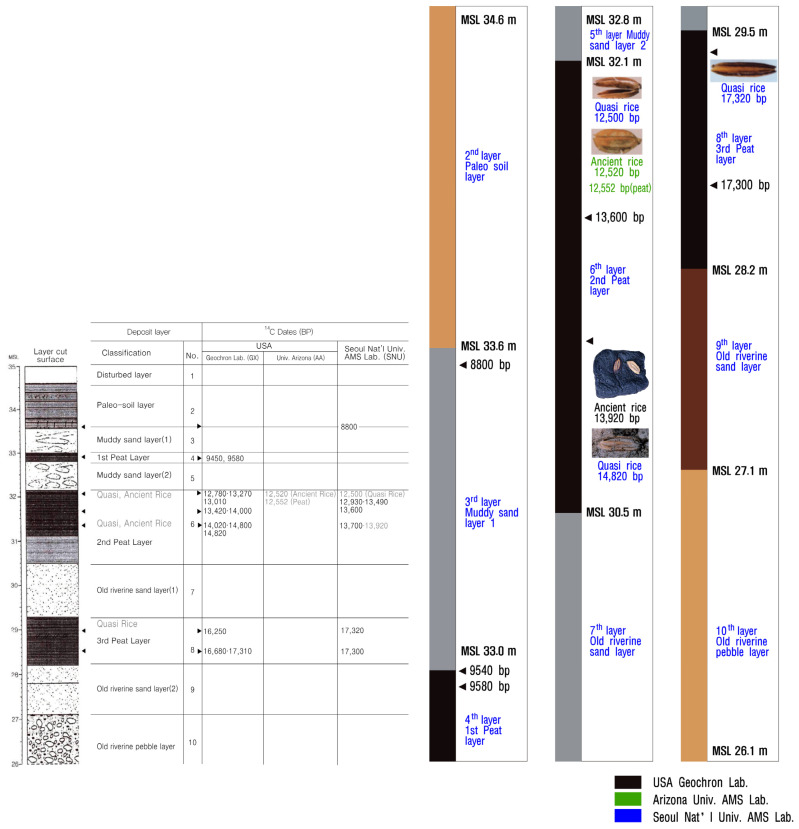
Cross-section of the excavated site at Sorori, Cheongju, in Korea. The ancient rice that was excavated was located at the following heights above sea level: Quasi-rice 3 (12,780 bp) about 32.0 m, ancient short rice seed 2 (13,490 bp) about 31.9 m, Quasi-rice seed 2 (14,820 bp) about 31.8 m, ancient short rice seed 1 (13,920 bp) about 31.1 m, and Quasi-rice seed 1 (17,310 bp) about 29.2 m [21].

**Figure 4 plants-13-01948-f004:**
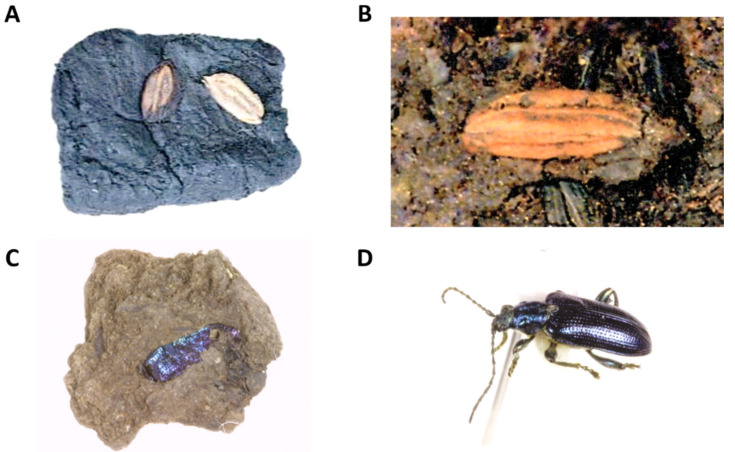
Carbonized rice and insect fossils excavated from Sorori, Cheongju. (**A**) Sorori ancient rice in situ; (**B**) Quasi-rice, in situ; (**C**) an example of the insect fossils found in the Sorori peat layer; (**D**) a present specimen of *Plateumaris sericea* (dark blue variation with violet reflection on wings and green reflection on legs).

**Figure 5 plants-13-01948-f005:**
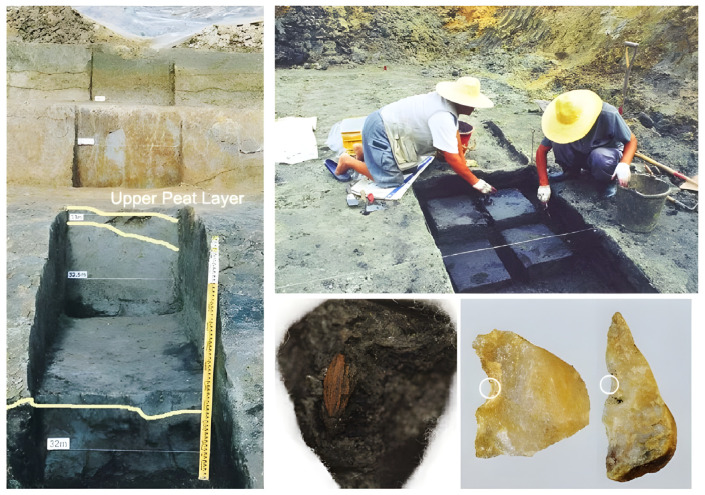
Sampling location in Sorori, rice sample, and a stone artifact that could have been used for cutting rice grains from rice plants in the rice field. The white circle means the cutting edge of notched tools.

**Figure 6 plants-13-01948-f006:**
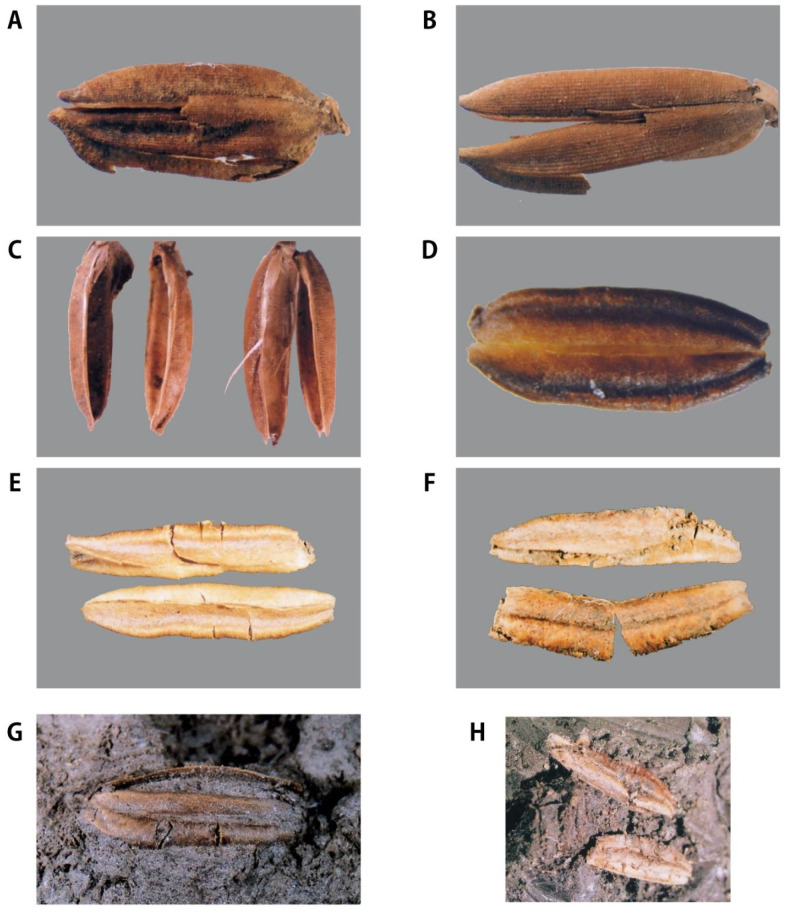
Comparison and analysis of the morphology of rice seeds excavated in Sorori. (**A**) Chaff of ancient short-grain rice (*japonica* type); (**B**) chaff of ancient long smooth-grain rice (*indica* type); (**C**) chaff of Quasi-rice I grain, half only; (**D**) chaff of complete Quasi-rice II grain; (**E**,**F**) chaff of Quasi-rice II grain, half only; (**G**) Quasi-rice II, complete at the time of excavation; (**H**) Quasi-rice II, incomplete at the time of excavation.

**Figure 7 plants-13-01948-f007:**
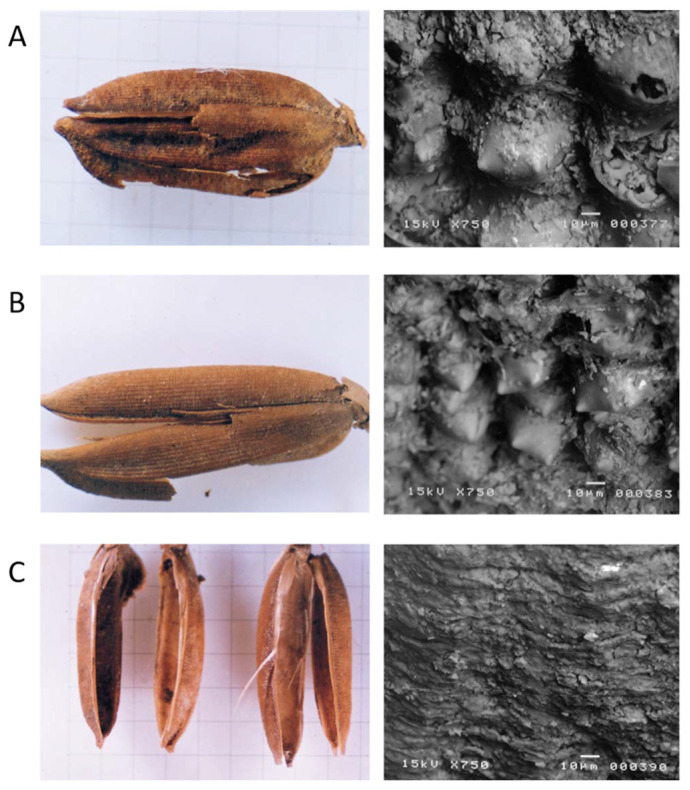
Analysis of the morphological characteristics of rice seeds excavated from Sorori using SEM. (**A**) Protuberance of ancient short rice grains (*japonica* type); (**B**) protuberance of ancient long smooth rice grains (*indica* type); (**C**) protuberance of Quasi-rice grains.

**Figure 8 plants-13-01948-f008:**
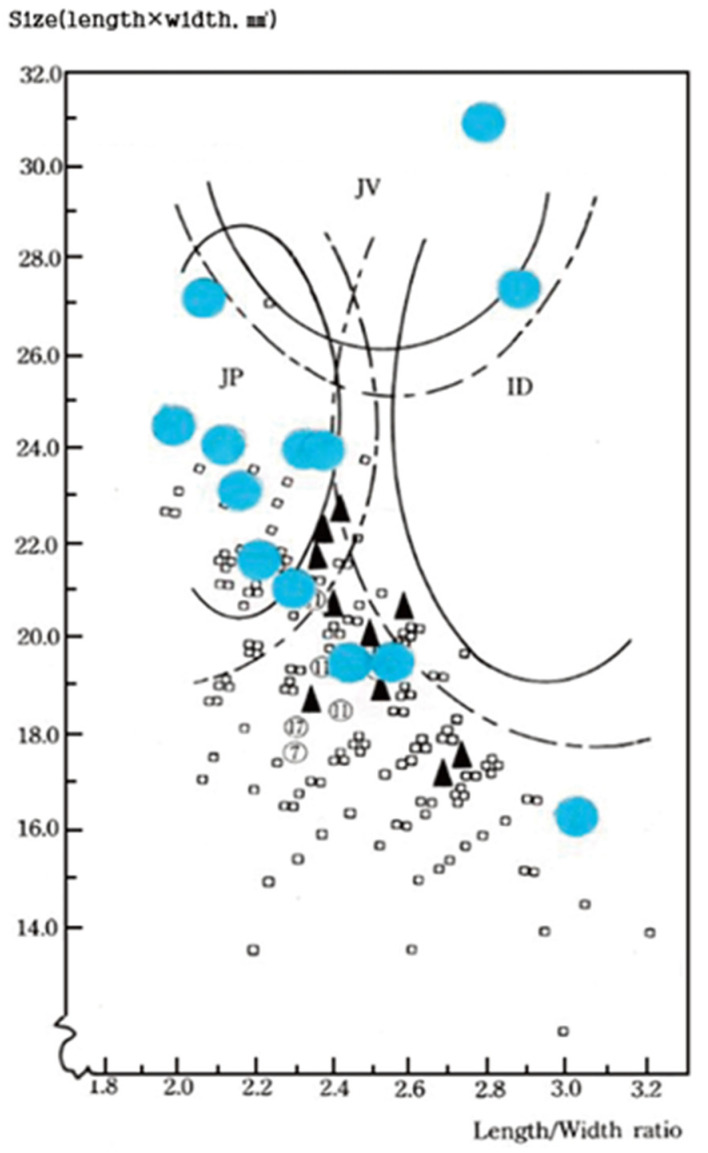
Rice grain distribution in association with grain size and length/width ratio from 200 grains collected Koyang city, and 13 ancient grains excavated at Sorori, Cheongju in Korea. ▲: Gawaji I (*n* = 10 about BP); □: Gawaji Π (*n* = 287 about BP); ●: Sorori (*n* = 13, 15,000 BP). The semi-circule represents a general grain size ratio range that includes seed length/width belonging to existing ecotypes, *indica* (ID), *japonica* (JP), and *javanica* (JV) [24].

**Figure 9 plants-13-01948-f009:**
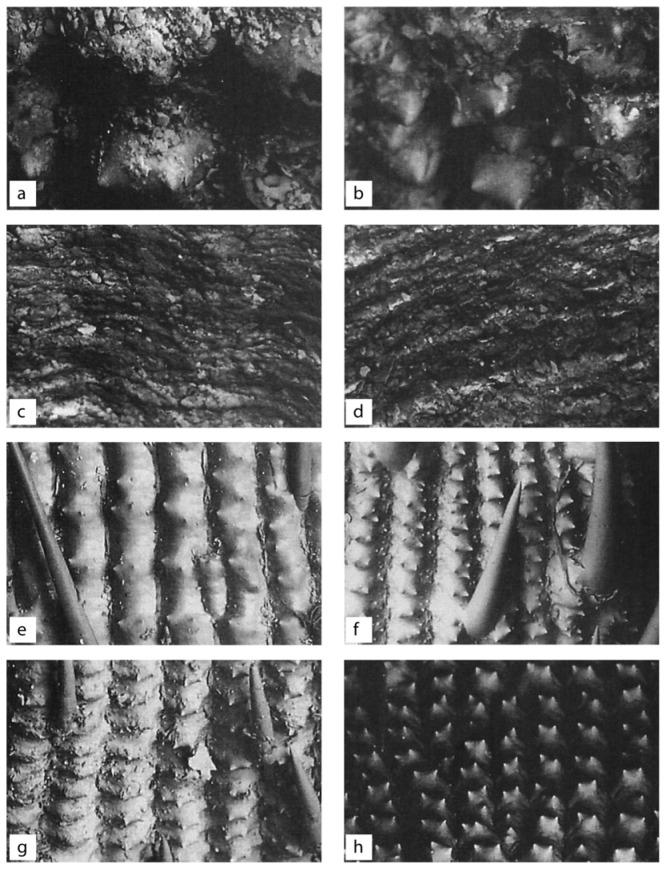
Bi-peak protuberances on the surface of glume obtained via SEM [9]. (**a**) Ancient rice: Short grain (**b**) Ancient rice: Long grain (**c**) Quasirice-1: No protuberance (**d**) Quasirice-2: No protuberance (**e**) Current cultivar: Hwasung-byo (**f**) Current cultivar: IR 36 (**g**) *O. glabrmima*: African rice (**h**) *O. rulfipogon*: Wild rice.

**Figure 10 plants-13-01948-f010:**
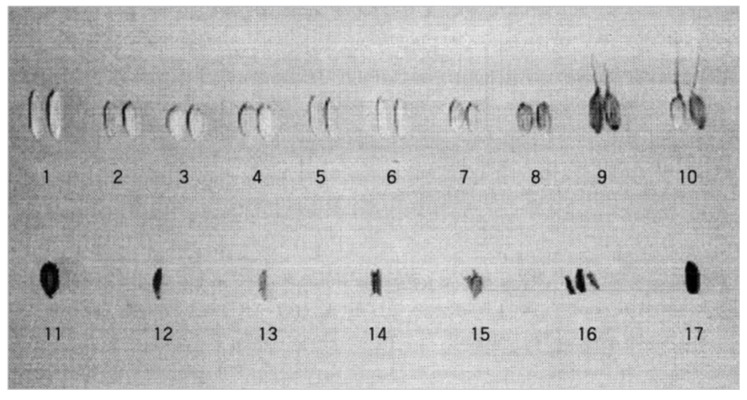
Appearance before the DNA extraction of cultivated rice, weedy rice, wild relatives of rice, carbonized rice excavated from Gawaji, Ilsan, and six types of ancient carbonized rice excavated from Sorori, Cheongju, Korea. 1: IR 36; 2: Pehku; 3: Nagdong; 4: Taichung 65; 5: Gyeongsan Aengmi 2; 6: Suncheon Aengmi 1; 7: Hapcheon Aengmi 3; 8: Seongju Aengmi 8; 9: W1944; 10: W130; 11: Gawaji excavated rice; 12: ancient short-grain; 13: Quasi-rice 1; 14: Quasi-rice 2; 15: ancient short-grain; 16: Quasi-rice 3; 17: Quasi-rice 4.

**Figure 11 plants-13-01948-f011:**
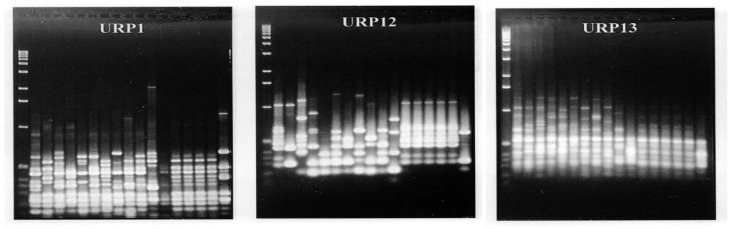
PCR band patterns of DNA extracted from rice husk using seven multi-range universal primers, URP1, URP12, and URP13.

**Figure 12 plants-13-01948-f012:**
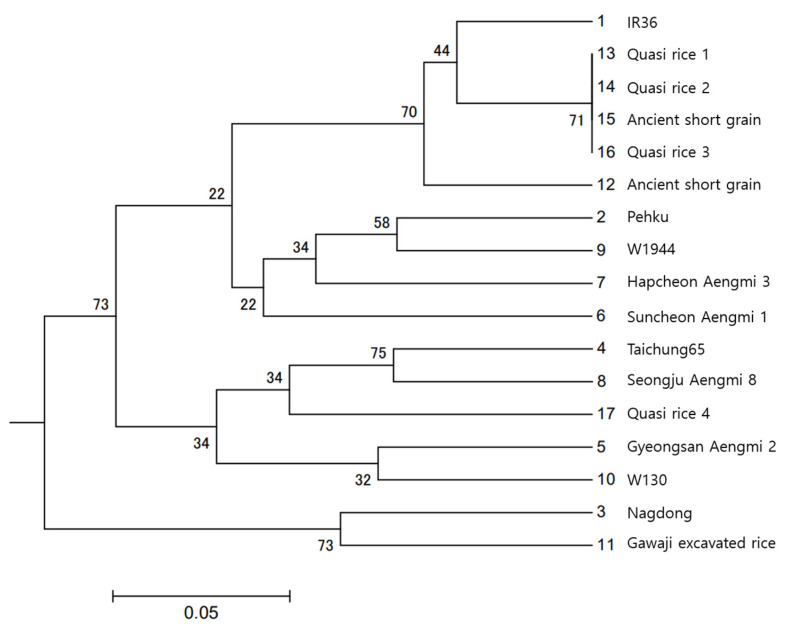
Phylogenetic analysis of ancient and current rice DNAs extracted from a single hull and amplified by universal rice primers, URP1, URP12, and URP13. The phylogenetic tree was constructed using the UPGMA method by POPTREE2 software with the UPGMA tree displayed using Nei’s standard genetic distance method.

**Figure 13 plants-13-01948-f013:**
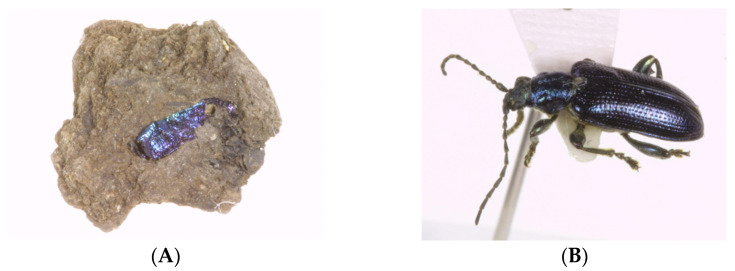
(**A**) Insect Fossil 1 excavated at Sorori (collection container number: 70); (**B**) example of extant species specimen—*Plateumaris sericea*, bronze color variant.

**Figure 14 plants-13-01948-f014:**
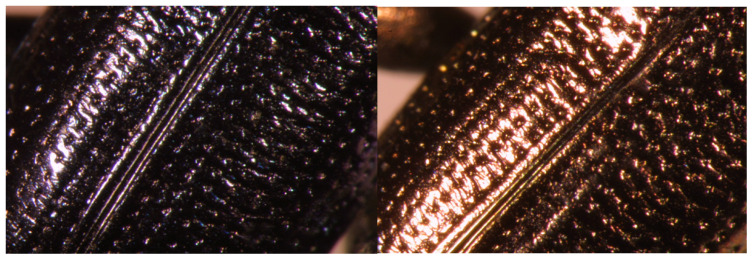
Examples of color variation in modern species of *Plateumaris sericea*.

**Figure 15 plants-13-01948-f015:**
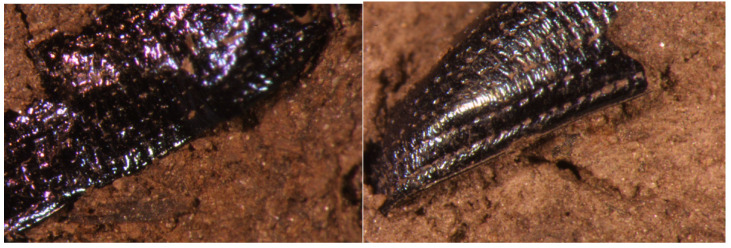
Color variation in the excavated fossil species *Plateumaris sericea*.

**Figure 16 plants-13-01948-f016:**
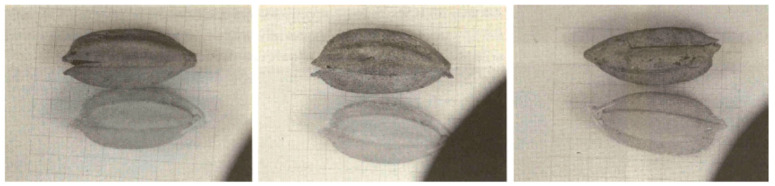
Sorori ancient rice grains found at second excavation in 2001.

**Figure 17 plants-13-01948-f017:**
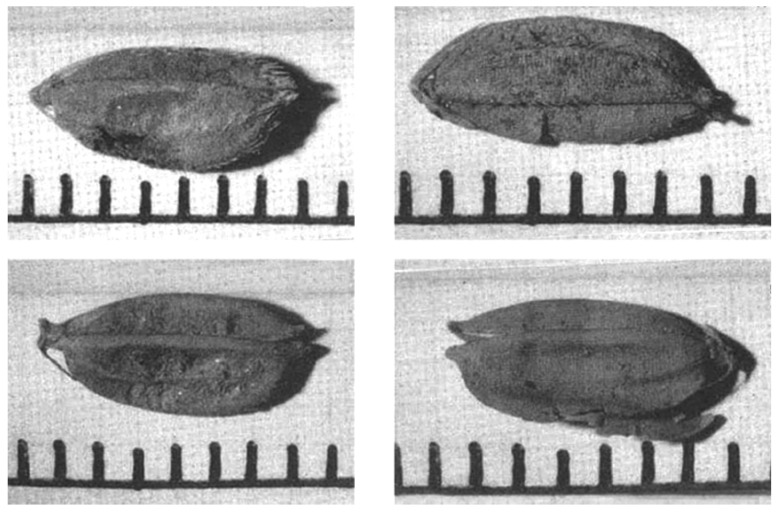
Ancient rice grains found at Ilsan excavation in 1991.

**Figure 18 plants-13-01948-f018:**
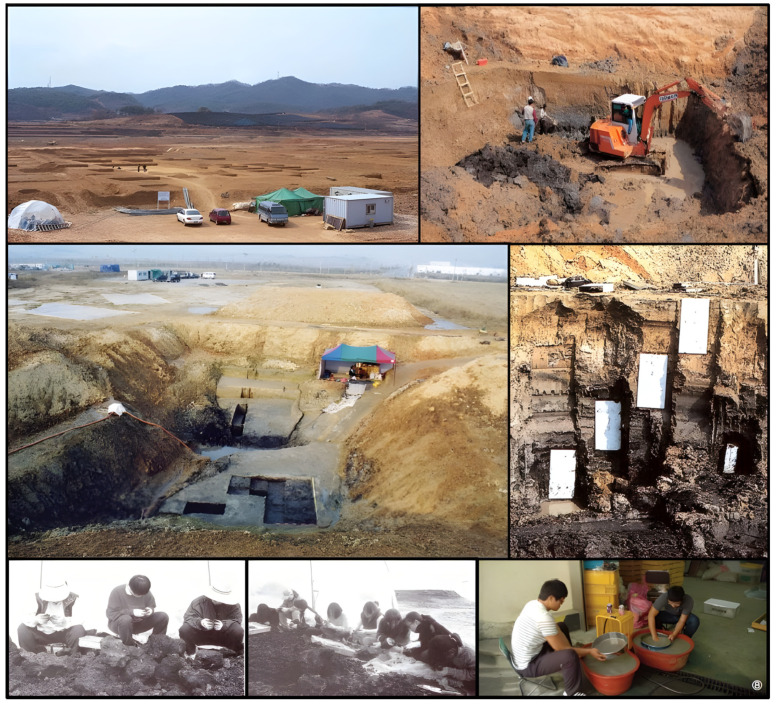
Scenes of excavation at Sorori site (A-II) in September 2001.

**Figure 19 plants-13-01948-f019:**
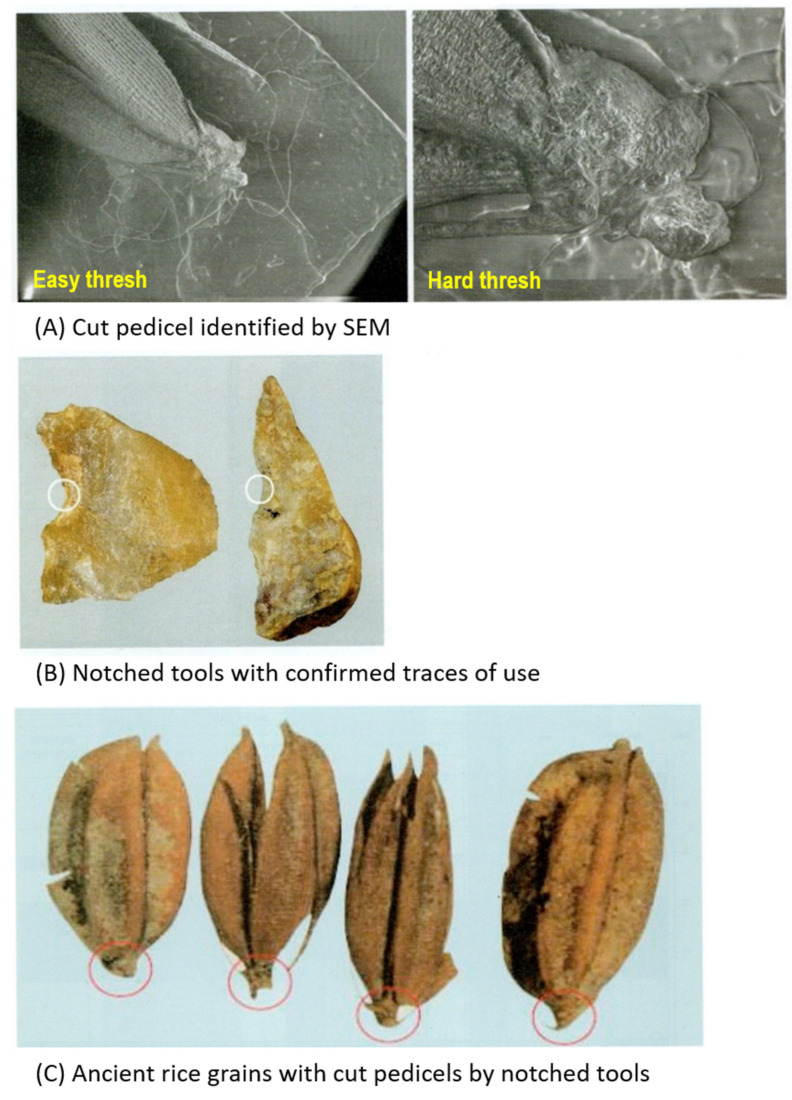
The excavated rice grains and stone artifacts that could have been used for cutting rice grains from rice plants in the rice field at Cheongju Sorori site. The white (**B**) and red (**C**) circles mean the cut part of stones and rice grains, respectively.

**Table 1 plants-13-01948-t001:** Radiation ages of Sorori rice from three AMS laboratories.

No.	Depth (MSL)	Deposit Layer	^14^C Date (BP)			
			Geochron (1998)	Geochron (2001)	SNU (2001)	NSF AZ (2009)
1	34.6–35	Disturbed layer				
2	33.6–34.6	Paleo-soil				
3	33–33.6	Muddy sand 1			8800 ± 90	
4	32.8–33	Upper peat		9450 ± 40		
				9580 ± 40		
5	32.1–32.8	Muddy sand 2				
6	30.5–32.1	Middle peat Quasi-rice and ancient rice were found in this layer	13,010 ± 190	12,780 ± 170–13,270 ± 180	12,500 ± 200	12,520 ± 150
			14,820 ± 250	13,420 ± 180	(quasi-rice)	(ancient rice)
				14,000 ± 190–14,020 ± 190	12,930 ± 400	12,552 ± 90
				14,800 ± 210	13,490 ± 150	(peat)
					13,600 ± 300	modern
					13,700 ± 200	(w/preservatives)
					13,920 ± 200	
7	29.3–30.5	Old riverine sand				
8	28.2–29.3	Lower peat	17,310 ± 310	16,250 ± 50	17,320 ± 200	
				16,680 ± 50	17,300 ± 150	
9	27.1–28.2	Old riverine sand				
10	26.0–27.1	Old riverine pebble				

**Table 2 plants-13-01948-t002:** Size of rice hulls found at Sorori site (28 May 1998).

Shape of Seed	No. of Sample	Length of Seed (mm)	Width of Seed (mm)	L/W Ratio *	Remarks
Rice (short)	A1-1	7.5	3.2	2.3	*japonica* type
	A1-2	6.9	2.8	2.5	
	A1-3	7.0	2.8	2.5	
	A1-4	6.2	2.6	2.4	
	A3-3	8.8	3.1	2.8	
	A3-4	7.0	2.3	3.0	
	A11-1	7.5	3.2	2.3	
	A11-2	7.1	3.4	2.1	
Rice (long)	A3-1	9.5	3.5	2.7	*indica* type
Quasi-rice I	A2-1	7.5	1.5	-	Half only ^†^
	A2-2	7.5	1.4		
	A2-3	7.5	1.5		
	A2-4	6.9	1.4		
	A2-5	8.5	1.8		
	A4-1	8.2	1.6		
Quasi-rice II	A4-2	7.1	2.0	-	Complete pair ^¶^
	A4-3	7.0	1.8		Half only
	A5-1	7.2	2.5		Complete pair
	A6-1	6.6	1.8		Half only
	A6-2	6.5	1.6		
	AA6-3	5.6	1.6		
	6-4	5.1	1.5		

* L/W, the length-to-width ratio of the seed; ^†^ half only, the shape of half an excavated rice seed; ^¶^ complete pair, the shape of a complete excavated rice seed.

**Table 3 plants-13-01948-t003:** Sorori ancient rice grains found at the second excavation.

No. of Sample	Length of Seed (mm)	Width of Seed (mm)	Remark
B1-1	7.0	3.3	*japonica* type
B1-2	7.5	3.6	
B1-3	7.0	3.5	
B2	6.8	3.2	
B3	6.0	1.1	Quasi-rice (half pair)
B4	6.0	1.7	
B5	7.4	1.6	
B6	6.5	2.2	
B7	7.2	1.8	
B8-1	6.6	1.6	
B8-2	6.5	1.8	
B8-3	5.7	1.4	
B9	8.5	1.7	
B10	6.5	1.3	
B11	5.0	1.9	
B12	5.3	1.3	
B13	5.2	1.4	
B14	8.0	1.6	

**Table 4 plants-13-01948-t004:** Carbonized rice hulls found at Ilsan site I.

No. of Sample	Length of Seed (mm)	Width of Seed (mm)		L/W Ratio	Remark
		Palea	Lemma		
I-1	6.87	2.31	0.91	2.12	Pubescence
I-2	6.45	2.24	1.00	1.99	Smooth
I-3	6.94	2.09	1.04	2.21	Pubescence
I-4	7.18	2.06	1.01	2.33	Pubescence
I-5	6.68	2.22	0.99	2.07	Smooth
I-6	7.04	2.05	0.97	2.33	Pubescence
I-7	6.77	2.05	0.96	2.25	Smooth
I-8	6.69	2.19	0.73	2.29	Smooth
I-9	6.53	broken	-	-	-
I-10	6.46	broken	-	-	-

**Table 5 plants-13-01948-t005:** Comparison of excavated rice hulls.

Site	Grain Size	Peat Layer Rice Hull Excavated		C-14 Age Years
	(Length)	From Surface	Sea Level	BP ± 1
Sorori 1st	6.2~8.8	2.9~3.6	32.2~32.1	13,010 ± 190
	9.5		31.4~31.3	14,820 ± 250
Sorori 2nd	6.8~7.5	2.9~3.6	32.2~32.1	12,930 ± 400
				13,920 ± 200
Ilsan	6.5~7.2	1	4.7~5/4	4070–5650
ck (Ilpumbyeo)	6.9 ± 0.34			

## Data Availability

Data is contained within the article.

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
