# Peer review of "The First Domesticated ‘Cheongju Sorori Rice’ Excavated in Korea"

_plants, 2024, doi:10.3390/plants13141948_

Round 1

Reviewer 1 Report

Comments and Suggestions for Authors

This paper deals with an extremely important and fascinating topic, delving as it does into the very birth of agriculture in Asia. However, it suffers from numerous flaws, and should be greatly revised before being resubmitted.

First of all, I would recommend a thorough checking of the English language, some statements are in their current form very difficult to understand.

Check that species names are italicized.

In Fig. 3 “Quasi” is misspelled as “Qausi”

Paragraphs 2.3.3. Morphological variations and 2.3.4. Genetic variation of grain size and length/width ratio should be rewritten, because as it stands they are of difficult understanding, due to typos and syntax errors. Also, I would not use the term “genetic” so liberally, when what the Authors have are just morphological differences. I suggest merging the two paragraphs and find some better way to display the observed variability than Fig. 8. Why a PCA or some other multivariate exploratory techniques was not used? In Fig. 8 grain size (measured how?) is shown as a function of length/width ratio, but no further explanation is provided. The same Figure is sloppily made, the huge blue dots of Sorori (I think) are indicated as black in the legend, and in the plot one can see unexplained circled numbers (7, 11, 17). Besides, what is the meaning of the arcs?

 The weakest part of the paper is the “genetic analysis”. First, I wish I could see pictures at higher resolution, because RAPDs are always tricky to score and need good quality gels and pictures.

Then, it must be borne to mind again that a RAPD band IS NOT a genetic marker, and surely NOT a gene, so that the statement “This result showed that 34.1% of genes detected in the four carbonized rice was same as those of the modern rice” is void.

Before using RAPD bands as genetic markers one should transform the data into allelic frequencies, considering that RAPD bands are inherited in a dominant way and using the appropriate corrections. One can use them as phenotypic markers, however, but no references to “genetic” must be allowed (in Table 5, the total number of bands in the last column is 75 and not 15).

The phylogenetic reconstruction is wanting, too. Here, correctly, the bands are treated as phenotypic markers, but many information is missing. First, one should clearly state which method was used to build the distance matrix using NTSys (some more recent software are available), then a dendrogram should always be validated by bootstrapping, which is notably missing from the analyses. Without this validation, one cannot be sure which nodes of the dendrogram are significant. Again, while I can understand running two different analyses for the “true” RAPD primers and the URP primers, I do not see the rationale of obtaining two separate phylogenies for URP1, URP12, and URP13 on one side and URP2, URP4, URP6, and URP15 on the other.

 In the (too) short Discussion I could not find any part about the “genetic analysis”, which left me surprised. Please discuss this in a future revision, taking into account the already made comments. Finally, the Discussion looks more like a retelling of the Results rather than a true discussion and should be completely rethought (no references to other results from the literature are present).

 The correct reference for the NJ method is not Nei 1987, rather Saitou & Nei 1987, that can be found here: https://doi.org/10.1093/oxfordjournals.molbev.a040454

Comments on the Quality of English Language

At the cost of repeating myself, please check the English throughout the whole ms.

Author Response

Response to Reviewer 1

This paper deals with an extremely important and fascinating topic, delving as it does into the very birth of agriculture in Asia. However, it suffers from numerous flaws, and should be greatly revised before being resubmitted.

First of all, I would recommend a thorough checking of the English language, some statements are in their current form very difficult to understand.

Check that species names are italicized.

-- Thank you for your kind comments. We changed all species in ithalics.

In Fig. 3 “Quasi” is misspelled as “Qausi”.

-- Thank you for your kind comments. Yes, three “Qausi” in Fig. 3 were mistakes. We have corrected them.

Paragraphs 2.3.3. Morphological variations and 2.3.4. Genetic variation of grain size and length/width ratio should be rewritten, because as it stands they are of difficult understanding, due to typos and syntax errors. Also, I would not use the term “genetic” so liberally, when what the Authors have are just morphological differences.

I suggest merging the two paragraphs and find some better way to display the observed variability than Fig. 8. Why a PCA or some other multivariate exploratory techniques was not used? In Fig. 8 grain size (measured how?) is shown as a function of length/width ratio, but no further explanation is provided. The same Figure is sloppily made, the huge blue dots of Sorori (I think) are indicated as black in the legend, and in the plot one can see unexplained circled numbers (7, 11, 17). Besides, what is the meaning of the arcs?

-- Thank you for your kind comments. We have checked and corrected to allow more understandable according to your comments as rewritten in lines 305-317 and lines 319-337. Also, we revised “genetic” to “morphological”.

-- We put two paragraphs into one as “2.3.3. Morphological variations of grain size and length/width ratio”. The length/width ratio was described in “Materials and Methods”.

-- The answer on “what is the meaning of the arcs?” is in lines 340-341. -- * The meaning of the semi-circular lines are as follow, JP: Japonica, ID: Indica, JV: Javanica.

 The weakest part of the paper is the “genetic analysis”. First, I wish I could see pictures at higher resolution, because RAPDs are always tricky to score and need good quality gels and pictures.

Then, it must be borne to mind again that a RAPD band IS NOT a genetic marker, and surely NOT a gene, so that the statement “This result showed that 34.1% of genes detected in the four carbonized rice was same as those of the modern rice” is void.

-- Thank you for your important points. Reviewer mentioned that “it must be borne to mind again that a RAPD band IS NOT a genetic marker, and surely NOT a gene, so that the statement”. However, I don not agree with “a RAPD band IS NOT a genetic marker”, because there so many cases that used RAPD as genetic markers as you can see on examples in the web page, https://www.ncbi.nlm.nih.gov/probe/docs/techrapd/ . (We can offer you more examples.) The researchers who analyzed RAPD bands as genetic markers used the bands when they showed Mendelian genetic function with repeated analysis.

-- Therefore, in PCR analysis in 1999, the 34.1% of the RAPD bands detected in the four carbonized rice were the same as modern rice. In fact, the RAPD analysis was done in 1999 when there was only limited technology was available. No more ancient DNA was available for the additional analysis since the very limited carbonized rice might be used for RAPD analysis.

Before using RAPD bands as genetic markers one should transform the data into allelic frequencies, considering that RAPD bands are inherited in a dominant way and using the appropriate corrections. One can use them as phenotypic markers, however, but no references to “genetic” must be allowed (in Table 5, the total number of bands in the last column is 75 and not 15).

-- Typically, the RAPD band is seen as codominant. The difference in RAPD band between carbonized and modern rice in our experiment can be said to be a difference in DNA sequences in the genomes of these rices. 

--- We modified the "genetic" as you pointed out with the “same pattern”. Table 5 is modified and supplemented.

The phylogenetic reconstruction is wanting, too. Here, correctly, the bands are treated as phenotypic markers, but many information is missing. First, one should clearly state which method was used to build the distance matrix using NTSys (some more recent software are available), then a dendrogram should always be validated by bootstrapping, which is notably missing from the analyses. Without this validation, one cannot be sure which nodes of the dendrogram are significant. Again, while I can understand running two different analyses for the “true” RAPD primers and the URP primers, I do not see the rationale of obtaining two separate phylogenies for URP1, URP12, and URP13 on one side and URP2, URP4, URP6, and URP15 on the other.

-- Thank you for your kind comments.  In 1999, Dr. Suh (who had passed away) and Dr. Cho used the NT-SYS program (Rohlf 2000). The analysis of the genetic similarity between modern rice and ancient rice seeds using the NT-SYS program (Rohlf 2000) was done based on “the presence or absence of bands on agarose gel of amplified PCR bands” obtained using URP1, URP12, and URP13 as shown in Figure 15. We are sorry that we cannot reconstruct the phylogenetic tree, because the raw data are not available now. As you know the carbonized rice hull samples are no longer available, because they were really original and unique samples in the world.

-- We deleted the description on URP2, URP4, URP6, and URP15 primers, and we kept description on the results with URP1, URP12, and URP13 primers.

 In the (too) short Discussion I could not find any part about the “genetic analysis”, which left me surprised. Please discuss this in a future revision, taking into account the already made comments. Finally, the Discussion looks more like a retelling of the Results rather than a true discussion and should be completely rethought (no references to other results from the literature are present).

-- Thank you for the reviewer's critical point. We have rewritten the Discussion including the description on the DNA analysis. Please check the Discussion.

 The correct reference for the NJ method is not Nei 1987, rather Saitou & Nei 1987, that can be found here: https://doi.org/10.1093/oxfordjournals.molbev.a040454

-- Thank you for the reviewer's valuable information. We have changed it according to the reviewer comments as follows:

26. N Saitou, M Nei, The neighbor-joining method: a new method for reconstructing phylogenetic trees., Molecular Biology and Evolution, Volume 4, Issue 4, Jul 1987: 406–425.

Reviewer 2 Report

Comments and Suggestions for Authors

I checked your manuscript and described comments below.

The origin of rice cultivation is very important in understanding the history of its spread.

This paper does a very good job of researching the origins of rice cultivation.

I think you should consider the following points.

1.       I have a suggestion for the molecular phylogenetic trees in Figures 15, 16, and 17. Recently, the mainstream method is to calculate the boot strap value using the maximum likelihood method and examine correlations. I think it would be better to include the boot strap value in these figures as well.

2.       It's not a big problem, but there are kanji in Ref. 14. I think it's better to write it in English.

I don't think this paper has various major mistakes or grammatical problems.

Author Response

Response to Reviewer 2

I checked your manuscript and described comments below.

The origin of rice cultivation is very important in understanding the history of its spread.

This paper does a very good job of researching the origins of rice cultivation.

I think you should consider the following points.

  1. I have a suggestion for the molecular phylogenetic trees in Figures 15, 16, and 17. Recently, the mainstream method is to calculate the boot strap value using the maximum likelihood method and examine correlations. I think it would be better to include the boot strap value in these figures as well.

-- Thank you for your kind comments.  In 1999, Dr. Suh (who had passed away) and Dr. Cho used the NT-SYS program (Rohlf 2000). The analysis of the genetic similarity between modern rice and ancient rice seeds using the NT-SYS program (Rohlf 2000) was done based on “the presence or absence of bands on agarose gel of amplified PCR bands” obtained using URP1, URP12, and URP13 as shown in Figure 15. We are sorry that we cannot reconstruct the phylogenetic tree, because the raw data are not available now. As you know the carbonized rice hull samples are no longer available, because they were really original and unique samples in the world.

-- We deleted the description on URP2, URP4, URP6, and URP15 primers, and we kept description on the results with URP1, URP12, and URP13 primers.

  1. It's not a big problem, but there are kanji in Ref. 14. I think it's better to write it in English.

I don't think this paper has various major mistakes or grammatical problems.

-- Thank you for your kind comments.

Reviewer 3 Report

Comments and Suggestions for Authors

 The manuscript titled “The Oldest ‘Cheongju Sorori Rice’ Excavated in Korea” is devoted to analysis of archaeological and other facts of ‘Cheongju Sorori Rice’ ancient cultivation in Korea. Excavation was carried out twice at the Sorori paleolithic site, Cheongju in the Republic of Korea. Rice seeds were excavated in 1998 and 2001.  Geologist and radiologist confirmed the peat layer which the rices were found that it goes back to older than 15,000 years.

In this study, the morphological characteristics, crushing, and RAPD band patterns according to the genetic polymorphism of rice grains in Cheongju Sorori were compared and analyzed for genetic similarities and differences with wild rice, weed rice, and modern rice. Based on the morphological, ecological, and physiological variations of rice grains excavated from the Soori site the Authors have suggested the origin of rice domestication in Korea.

The manuscript describes a lot of data, but, the Authors have mixed two different approaches – literature review and experimental report. Unfortunately, RAPD data alone, without support from other molecular data (DNA sequencing), cannot be accepted as reliable support of the conclusions made in the Manuscript.

Comments on the Quality of English Language

Extensive editing of English language required

Author Response

Reviewer 3

The manuscript titled “The Oldest ‘Cheongju Sorori Rice’ Excavated in Korea” is devoted to analysis of archaeological and other facts of ‘Cheongju Sorori Rice’ ancient cultivation in Korea. Excavation was carried out twice at the Sorori paleolithic site, Cheongju in the Republic of Korea. Rice seeds were excavated in 1998 and 2001.  Geologist and radiologist confirmed the peat layer which the rices were found that it goes back to older than 15,000 years.

In this study, the morphological characteristics, crushing, and RAPD band patterns according to the genetic polymorphism of rice grains in Cheongju Sorori were compared and analyzed for genetic similarities and differences with wild rice, weed rice, and modern rice. Based on the morphological, ecological, and physiological variations of rice grains excavated from the Soori site the Authors have suggested the origin of rice domestication in Korea.

The manuscript describes a lot of data, but, the Authors have mixed two different approaches – literature review and experimental report. Unfortunately, RAPD data alone, without support from other molecular data (DNA sequencing), cannot be accepted as reliable support of the conclusions made in the Manuscript.

-- Thank you for your kind comments. Yes, your comment is reasonable. We agree that the estimation of the origin of early domesticated rice should be further confirmed by the comparative analysis based on DNA sequencing of the excavated rice and the current rice. We are working on it now.

Round 2

Reviewer 1 Report

Comments and Suggestions for Authors

Apparently, the Authors did not take into consideration any of my comments about the genetics and others. See below

I suggest merging the two paragraphs and find some better way to display the observed variability than Fig. 8. Why a PCA or some other multivariate exploratory techniques was not used? In Fig. 8 grain size (measured how?) is shown as a function of length/width ratio, but no further explanation is provided. The same Figure is sloppily made, the huge blue dots of Sorori (I think) are indicated as black in the legend, and in the plot one can see unexplained circled numbers (7, 11, 17). Besides, what is the meaning of the arcs?

-- Thank you for your kind comments. We have checked and corrected to allow more understandable according to your comments as rewritten in lines 305-317 and lines 319-337. Also, we revised “genetic” to “morphological”.

-- We put two paragraphs into one as “2.3.3. Morphological variations of grain size and length/width ratio”. The length/width ratio was described in “Materials and Methods”.

-- The answer on “what is the meaning of the arcs?” is in lines 340-341. -- * The meaning of the semi-circular lines are as follow, JP: Japonica, ID: Indica, JV: Javanica.

This is not what I meant, the “meaning” of the lines is not JP, ID or JV, but how they separate JP, JD and JV: are those lines confidence intervals or what? How have they been estimated?

The weakest part of the paper is the “genetic analysis”. First, I wish I could see pictures at higher resolution, because RAPDs are always tricky to score and need good quality gels and pictures.

Then, it must be borne to mind again that a RAPD band IS NOT a genetic marker, and surely NOT a gene, so that the statement “This result showed that 34.1% of genes detected in the four carbonized rice was same as those of the modern rice” is void.

-- Thank you for your important points. Reviewer mentioned that “it must be borne to mind again that a RAPD band IS NOT a genetic marker, and surely NOT a gene, so that the statement”. However, I don not agree with “a RAPD band IS NOT a genetic marker”, because there so many cases that used RAPD as genetic markers as you can see on examples in the web page, https://www.ncbi.nlm.nih.gov/probe/docs/techrapd/ . (We can offer you more examples.) The researchers who analyzed RAPD bands as genetic markers used the bands when they showed Mendelian genetic function with repeated analysis.

-- Therefore, in PCR analysis in 1999, the 34.1% of the RAPD bands detected in the four carbonized rice were the same as modern rice. In fact, the RAPD analysis was done in 1999 when there was only limited technology was available. No more ancient DNA was available for the additional analysis since the very limited carbonized rice might be used for RAPD analysis.

Before using RAPD bands as genetic markers one should transform the data into allelic frequencies, considering that RAPD bands are inherited in a dominant way and using the appropriate corrections. One can use them as phenotypic markers, however, but no references to “genetic” must be allowed (in Table 5, the total number of bands in the last column is 75 and not 15).

-- Typically, the RAPD band is seen as codominant. The difference in RAPD band between carbonized and modern rice in our experiment can be said to be a difference in DNA sequences in the genomes of these rices.

--- We modified the "genetic" as you pointed out with the “same pattern”. Table 5 is modified and supplemented.

The Authors’ reply is wrong. This is probably due to the long time elapsed since RAPD markers were invented, so that their pitfalls (not their power) have been forgotten. However, I have been working on RAPDs since their invention and in close contact with the inventors themselves; I must therefore say that I am qualified to speak about them. Yes, a RAPD band is not a genetic marker, because for stating this a validation must occur, as example, as the Authors rightly say, by demonstrating its Mendelian inheritance (not function). In fact, a RAPD band can be many things, from the overlapping of different amplification products to an artefact (much common when dealing with RAPDs). Then, the Authors are again very wrong when they say: “Typically, the RAPD band is seen as codominant”; no, RAPDs are dominant markers, as the technique itself and thirty years of publications testify.

You can assume that your bands are genetic markers, when PCRs are of good quality, and repeatability of the data has been tested, but then you must transform your data, with the appropriate corrections, into allelic frequencies, because of dominance. What you can be reasonably sure about is that the absence of a RAPD band indicates a recessive homozygote aa, but the presence of a band can indicate both a AA and an Aa genotype. Last, RAPD means Random Amplified Polymorphic DNA, with focus on Random: even assuming that a band is a genetic marker, then the probability it is the amplification product of a part of a gene is very low indeed, so that, again, the Authors are not allowed to state that, because 34.1% of the bands are shared between modern and archaic rice, “This result showed that 34.1% of genes detected in the four carbonized rice was same as those of the modern rice”.

For a complete discussion about the above, and the correct treatment to be applied to RAPD data, I suggest reading Lynch and Milligan, Molecular Ecology 3: 91–99; 1994.

The phylogenetic reconstruction is wanting, too. Here, correctly, the bands are treated as phenotypic markers, but many information is missing. First, one should clearly state which method was used to build the distance matrix using NTSys (some more recent software are available), then a dendrogram should always be validated by bootstrapping, which is notably missing from the analyses. Without this validation, one cannot be sure which nodes of the dendrogram are significant. Again, while I can understand running two different analyses for the “true” RAPD primers and the URP primers, I do not see the rationale of obtaining two separate phylogenies for URP1, URP12, and URP13 on one side and URP2, URP4, URP6, and URP15 on the other.

-- Thank you for your kind comments. In 1999, Dr. Suh (who had passed away) and Dr. Cho used the NT-SYS program (Rohlf 2000). The analysis of the genetic similarity between modern rice and ancient rice seeds using the NT-SYS program (Rohlf 2000) was done based on “the presence or absence of bands on agarose gel of amplified PCR bands” obtained using URP1, URP12, and URP13 as shown in Figure 15. We are sorry that we cannot reconstruct the phylogenetic tree, because the raw data are not available now. As you know the carbonized rice hull samples are no longer available, because they were really original and unique samples in the world.

-- We deleted the description on URP2, URP4, URP6, and URP15 primers, and we kept description on the results with URP1, URP12, and URP13 primers.

I am sorry to hear of Dr. Su’s passing away, and I understand why many information is missing. However, there are still some points bothering me about the trees; in Figure 12 there is a bar labelled “Coefficient” with a scale from 0.8 to 0.4 decreasing towards the OTUs. Please specify what coefficient. A similar bar appears again in Figures 15 and 16, without a label, and with a scale from 0.5 to 1.0 increasing towards the OTUs. Please again specify the meaning of the scale; my guess is that it is the “similarity” cited in the text, but what kind of similarity index was used should be stated for a better understanding of the UPGMA trees.

A final afterthought: because of the lack of a bootstrap, in all trees the phylogenetic analysis is incomplete. Taking this into consideration, I would retain only the results shown in in Fig. 12 and in Fig. 16, where, the clustering of “ancient” samples away from modern rice is much clearer. The lack of a bootstrap analysis should however be noted.

In the (too) short Discussion I could not find any part about the “genetic analysis”, which left me surprised. Please discuss this in a future revision, taking into account the already made comments. Finally, the Discussion looks more like a retelling of the Results rather than a true discussion and should be completely rethought (no references to other results from the literature are present).

-- Thank you for the reviewer's critical point. We have rewritten the Discussion including the description on the DNA analysis. Please check the Discussion.

I appreciate the effort done by the Authors, but the point I raised has not been tackled; the Discussion is still much a repetition of the results. As example, when describing the morphological variability, why not to make references to the degree of variation observed among modern rice varieties?

Comments on the Quality of English Language

Despite some editing done, the English language should be thoroughly checked

Author Response

Response to Reviewer 1 _ R2

Apparently, the Authors did not take into consideration any of my comments about the genetics and others. See below

I suggest merging the two paragraphs and find some better way to display the observed variability than Fig. 8. Why a PCA or some other multivariate exploratory techniques was not used? In Fig. 8 grain size (measured how?) is shown as a function of length/width ratio, but no further explanation is provided. The same Figure is sloppily made, the huge blue dots of Sorori (I think) are indicated as black in the legend, and in the plot one can see unexplained circled numbers (7, 11, 17). Besides, what is the meaning of the arcs?

-- Thank you for your kind comments. We have checked and corrected to allow more understandable according to your comments as rewritten in lines 305-317 and lines 319-337. Also, we revised “genetic” to “morphological”.

-- We put two paragraphs into one as “2.3.3. Morphological variations of grain size and length/width ratio”. The length/width ratio was described in “Materials and Methods”.

-- The answer on “what is the meaning of the arcs?” is in lines 340-341. -- * The meaning of the semi-circular lines is as follow, JP: Japonica, ID: Indica, JV: Javanica.

This is not what I meant, the “meaning” of the lines is not JP, ID or JV, but how they separate JP, JD and JV: are those lines confidence intervals or what? How have they been estimated?

-- Thank you for your kind comments. Dr. Tae-Sik Park (who had passed away) draw the Figure 8. So we could not check the way how to draw the arcs for JP, ID, JV, but we have tried to figure it out what they are. But we were able to cite the following report. Here, we show you the part related to “the way how to draw the arcs for JP, ID, JV”.

According to the report, ‘Goyang Gawaji Carbonized Rice Seeds (I): Investigation and Research’ by Yung-Jo Lee and Tae-Sik Park (2014):

“Among the 300 carbonized rice seeds excavated in Gawaji, 287 measurable rice seeds were measured, and the length was 6.0~7.8 mm, with an average of 6.75 mm, and the width was 2.1~3.5 mm, with an average of 2.84 mm. Looking at the relationship between length and width, it could be seen that it was spread over a very wide area. In addition, the length/width ratio, which measures the size of rice seeds, averaged 2.39 mm, which was slightly thinner and longer than today's cultivated rice.

Generally, when classifying subspecies, they are classified according to length-to-width ratio and size distribution. Even if we assume that the Gawaji rice seeds have been buried in peat for a long time and thus might have changed in size, we can see that the distribution area in length and width is very wide. This fact shows that the rice seeds of Gawaji District 2 possess a wide variety of genetic traits from a rice breeding perspective, and most of them show characteristics of single-grain type (japonica), but compared to today's indica, it has 35 points. Since it belongs to the long-grained type (indica), it seemed to retain indica characteristics. The proportion of indica in the total was 12.3%...”

In the study on Goyang Kawaji carbonized rice seeds,

The semicircle that divides ecotypes in the drawing arcs of rice subspecies classification studied on rice seeds excavated from Districts 1 and 2 is not a classification line based on the results of any analysis, but is a general grain size ratio range that includes seed length/width belonging to existing ecotypes, indica(ID), japonica(JP) and javanica(JV). It is judged to have been drawn arbitrarily with reference to (Figure 70)( ‘Goyang Gawaji Carbonized Rice Seeds (I): Investigation and Research’ by Yung-Jo Lee and Tae-Sik Park, 2014). This is presumed to be the result of confirming which ecotype it belongs to by substituting the length/width ratio of rice seeds excavated from Sorori on a plane again in Figure 70.

The weakest part of the paper is the “genetic analysis”. First, I wish I could see pictures at higher resolution, because RAPDs are always tricky to score and need good quality gels and pictures.

Then, it must be borne to mind again that a RAPD band IS NOT a genetic marker, and surely NOT a gene, so that the statement “This result showed that 34.1% of genes detected in the four carbonized rice was same as those of the modern rice” is void.

-- Thank you for your important points. Reviewer mentioned that “it must be borne to mind again that a RAPD band IS NOT a genetic marker, and surely NOT a gene, so that the statement”. However, I do not agree with “a RAPD band IS NOT a genetic marker”, because there so many cases that used RAPD as genetic markers as you can see on examples in the web page, https://www.ncbi.nlm.nih.gov/probe/docs/techrapd/. (We can offer you more examples.) The researchers who analyzed RAPD bands as genetic markers used the bands when they showed Mendelian genetic function with repeated analysis.

-- Therefore, in PCR analysis in 1999, the 34.1% of the RAPD bands detected in the four carbonized rice were the same as modern rice. In fact, the RAPD analysis was done in 1999 when there was only limited technology was available. No more ancient DNA was available for the additional analysis since the very limited carbonized rice might be used for RAPD analysis.

Before using RAPD bands as genetic markers one should transform the data into allelic frequencies, considering that RAPD bands are inherited in a dominant way and using the appropriate corrections. One can use them as phenotypic markers, however, but no references to “genetic” must be allowed (in Table 5, the total number of bands in the last column is 75 and not 15).

-- Typically, the RAPD band is seen as codominant. The difference in RAPD band between carbonized and modern rice in our experiment can be said to be a difference in DNA sequences in the genomes of these rices.

--- We modified the "genetic" as you pointed out with the “same pattern”. Table 5 is modified and supplemented.

The Authors’ reply is wrong. This is probably due to the long time elapsed since RAPD markers were invented, so that their pitfalls (not their power) have been forgotten. However, I have been working on RAPDs since their invention and in close contact with the inventors themselves; I must therefore say that I am qualified to speak about them. Yes, a RAPD band is not a genetic marker, because for stating this a validation must occur, as example, as the Authors rightly say, by demonstrating its Mendelian inheritance (not function). In fact, a RAPD band can be many things, from the overlapping of different amplification products to an artefact (much common when dealing with RAPDs). Then, the Authors are again very wrong when they say: “Typically, the RAPD band is seen as codominant”; no, RAPDs are dominant markers, as the technique itself and thirty years of publications testify.

You can assume that your bands are genetic markers, when PCRs are of good quality, and repeatability of the data has been tested, but then you must transform your data, with the appropriate corrections, into allelic frequencies, because of dominance. What you can be reasonably sure about is that the absence of a RAPD band indicates a recessive homozygote aa, but the presence of a band can indicate both a AA and an Aa genotype. Last, RAPD means Random Amplified Polymorphic DNA, with focus on Random: even assuming that a band is a genetic marker, then the probability it is the amplification product of a part of a gene is very low indeed, so that, again, the Authors are not allowed to state that, because 34.1% of the bands are shared between modern and archaic rice, “This result showed that 34.1% of genes detected in the four carbonized rice was same as those of the modern rice”.

-- Thank you for your kind comments. According to the reviewers comments, we decided to delete the PCR results derived from RAPD primers on pages 12~15, “2.3.5. DNA variation of the carbonized rice, Experiment 1:”. And we reanalyzed the results derived from the Universal Rice Primers (URP primers) on page 15~17 using the UPGMA method of POPTREE2 software (Takezaki 2009) as shown in Figure 13. And then we rewrote the section based on new analysis in pages 13~14 and 23. And Figure 16 was also deleted.

For a complete discussion about the above, and the correct treatment to be applied to RAPD data, I suggest reading Lynch and Milligan, Molecular Ecology 3: 91–99; 1994.

-- Thank you for your kind comments. As you recommend the paper to me, I read it by Lynch and Milligan. I agree with their suggestion as follows, 'Due to the need for pruning loci with low-frequency null alleles, more loci also need to be sampled with RAPDs than with more conventional markers, and some problems of bias cannot be completely eliminated'. But I’d like to tell you that there was a limitation to carry out experiments with many samples and the repeated experiments due to short of the ancient carbonized samples that contained only very little limited rice DNA allowed to use.

Anyway, we decided to delete the PCR results derived from RAPD primers on pages 12~15, in “2.3.5. DNA variation of the carbonized rice, Experiment 1:”.

The phylogenetic reconstruction is wanting, too. Here, correctly, the bands are treated as phenotypic markers, but many information is missing. First, one should clearly state which method was used to build the distance matrix using NTSys (some more recent software are available), then a dendrogram should always be validated by bootstrapping, which is notably missing from the analyses. Without this validation, one cannot be sure which nodes of the dendrogram are significant. Again, while I can understand running two different analyses for the “true” RAPD primers and the URP primers, I do not see the rationale of obtaining two separate phylogenies for URP1, URP12, and URP13 on one side and URP2, URP4, URP6, and URP15 on the other.

-- Thank you for your kind comments. In 1999, Dr. Suh (who had passed away) and Dr. Cho used the NT-SYS program (Rohlf 2000). The analysis of the genetic similarity between modern rice and ancient rice seeds using the NT-SYS program (Rohlf 2000) was done based on “the presence or absence of bands on agarose gel of amplified PCR bands” obtained using URP1, URP12, and URP13 as shown in Figure 15. We are sorry that we cannot reconstruct the phylogenetic tree, because the raw data are not available now. As you know the carbonized rice hull samples are no longer available, because they were really original and unique samples in the world.

-- We deleted the description on URP2, URP4, URP6, and URP15 primers, and we kept description on the results with URP1, URP12, and URP13 primers.

I am sorry to hear of Dr. Suh’s passing away, and I understand why many information is missing. However, there are still some points bothering me about the trees; in Figure 12 there is a bar labelled “Coefficient” with a scale from 0.8 to 0.4 decreasing towards the OTUs. Please specify what coefficient. A similar bar appears again in Figures 15 and 16, without a label, and with a scale from 0.5 to 1.0 increasing towards the OTUs. Please again specify the meaning of the scale; my guess is that it is the “similarity” cited in the text, but what kind of similarity index was used should be stated for a better understanding of the UPGMA trees.

A final afterthought: because of the lack of a bootstrap, in all trees the phylogenetic analysis is incomplete. Taking this into consideration, I would retain only the results shown in in Fig. 12 and in Fig. 16, where, the clustering of “ancient” samples away from modern rice is much clearer. The lack of a bootstrap analysis should however be noted.

-- Thank you for your kind comments. According to the reviewers comments, we decided to delete the PCR results derived from RAPD primers on pages 12~15, in “2.3.5. DNA variation of the carbonized rice, Experiment 1”. And we reanalyzed the results derived from the Universal Rice Primers (URP primers) on page 15~17 using the UPGMA method of POPTREE2 software (Takezaki 2009) as shown in Figure 13. And then we rewrote the section based on new analysis in pages 13~14 and 23. And Figure 16 was also deleted.

In the (too) short Discussion I could not find any part about the “genetic analysis”, which left me surprised. Please discuss this in a future revision, taking into account the already made comments. Finally, the Discussion looks more like a retelling of the Results rather than a true discussion and should be completely rethought (no references to other results from the literature are present).

-- Thank you for the reviewer's critical point. We have rewritten the Discussion including the description on the DNA analysis. Please check the Discussion.

I appreciate the effort done by the Authors, but the point I raised has not been tackled; the Discussion is still much a repetition of the results. As example, when describing the morphological variability, why not to make references to the degree of variation observed among modern rice varieties?

-- Thank you for your kind comments. The Discussion has been rewritten based on the reviewers comments.

Reviewer 3 Report

Comments and Suggestions for Authors

The manuscript titled “The Oldest ‘Cheongju Sorori Rice’ Excavated in Korea” is devoted to analysis of archaeological and other facts of ‘Cheongju Sorori Rice’ ancient cultivation in Korea. Excavation was carried out twice at the Sorori paleolithic site, Cheongju in the Republic of Korea. Rice seeds were excavated in 1998 and 2001.  This fact has been previously reported, including paper of 2013 (Kim, K. J., Lee, Y. J., Woo, J. Y., & Jull, A. J. T. (2013). Radiocarbon ages of Sorori ancient rice of Korea. Nuclear Instruments and Methods in Physics Research, Section B: Beam Interactions with Materials and Atoms, 294, 675-679. https://doi.org/10.1016/j.nimb.2012.09.026). This report (Kim et al. 2013) stated that "Samples of Sorori ancient rice were excavated in 1998 from the Sorori Paleolithic site. Both ancient rice samples and surrounded peat from the Sorori site were dated. The AMS results confirmed that the ages of the rice and peat soil were 12,520 ± 150 and 12,552 ± 90 BP, respectively. These radiocarbon ages are consistent with the previously published data of quasi rice measured at Seoul National University ".

In this manuscript, Authors suggest that older peat soil with rice seed indicate older seeds age, but do not confirm the equal age of carbonized rice and peat.

Jing  et al. (2023) (Jing, CY., Zhang, FM., Wang, XH. et al. Multiple domestications of Asian rice. Nat. Plants 9, 1221–1235 (2023). https://doi.org/10.1038/s41477-023-01476-z) suggested that there are “… two leading hypotheses: a single domestication event in China or multiple domestication events in geographically separate areas. These two hypotheses differ in their predicted history of genes/alleles selected during domestication..” “Analysis of dataset of 1,578 resequenced genomes and 993 selected genes that generated phylogenetic trees on which japonica and indica formed a monophyletic group, suggesting that the domestication alleles of these genes originated only once in either japonica or indica.”

“Importantly, the domestication alleles of most selected genes (~80%) stemmed from wild rice in China, but the domestication alleles of a substantial minority of selected genes (~20%) originated from wild rice in South and Southeast Asia, demonstrating separate domestication events of Asian rice”.

In comparison to these and other research works that employed vast amount of genetic and phenetic data, this research reports besides the morphological characteristics, only RAPD band patterns to assay the genetic polymorphism of rice grains found in Cheongju Sorori and compare it with wild rice, weed rice, and modern rice. Based on these results, the Authors have suggested the origin of rice domestication in Korea.

Reviewer cannot accept the reported here RAPD results as reliable for several reasons:

1)         DNA isolated from the ancient carbonized rice samples has very high risk (close to 100%) of contamination by microbial and fungal DNA of both old and more recent age. DNA sequencing must be applied to confirm the identity of these DNA samples.

2)         The resulting RAPD profiles (Fig. 11) yielded very few diverse bands for the ancient samples (31 specific bands in total, Table 4). In opposite, Figure 14 show nearly identical RAPD (7 primers) profiles for ancient samples of diverse origin – lines 11: Gawaji excavated rice, 12: Ancient short grain, 13: Quasi rice 1, 14: Quasi rice 1, 15: Ancient short grain, and 16: Quasi rice 2. At the same time, modern rice samples were variable. This fact has no any reasonable explanation.

3)         Reliability of the phylogenetic constructions shown in Figure 11, 15-17 cannot be evaluated because they are shown without any bootstrap of another statistical analysis. 

The manuscript describes a lot of data, but, the particular  RAPD results cannot be accepted for publication.

It is highly recommended to remove this part from the manuscript.

Author Response

Response to Reviewer 3 _ R2

The manuscript titled “The Oldest ‘Cheongju Sorori Rice’ Excavated in Korea” is devoted to analysis of archaeological and other facts of ‘Cheongju Sorori Rice’ ancient cultivation in Korea. Excavation was carried out twice at the Sorori paleolithic site, Cheongju in the Republic of Korea. Rice seeds were excavated in 1998 and 2001.  This fact has been previously reported, including paper of 2013 (Kim, K. J., Lee, Y. J., Woo, J. Y., & Jull, A. J. T. (2013). Radiocarbon ages of Sorori ancient rice of Korea. Nuclear Instruments and Methods in Physics Research, Section B: Beam Interactions with Materials and Atoms, 294, 675-679. https://doi.org/10.1016/j.nimb.2012.09.026). This report (Kim et al. 2013) stated that "Samples of Sorori ancient rice were excavated in 1998 from the Sorori Paleolithic site. Both ancient rice samples and surrounded peat from the Sorori site were dated. The AMS results confirmed that the ages of the rice and peat soil were 12,520 ± 150 and 12,552 ± 90 BP, respectively. These radiocarbon ages are consistent with the previously published data of quasi rice measured at Seoul National University ".

In this manuscript, Authors suggest that older peat soil with rice seed indicate older seeds age, but do not confirm the equal age of carbonized rice and peat.

-- Thank you for your comments. Yes, it is already confirmed that the carbonized rice and the peat have an equal age. Ancient rice and quasi rice were excavated from Sorori Site within 32.17~32.06m MSL (mean sea level), and four dating laboratories (GX, AA, SNU, KIGAM) identified peat (12,552 BP) and ancient rice (12,520 BP) and quasi rice (12,500 BP) were obtained. All experts (archaeologists, dating scientists, Quarternary geologist, etc.) agree that they were excavated from a very stable stratum. Out of 15,000 years, the 30~50 years difference could be regarded as the equal age in radiation dating.

Jing et al. (2023) (Jing, CY., Zhang, FM., Wang, XH. et al. Multiple domestications of Asian rice. Nat. Plants 9, 1221–1235 (2023). https://doi.org/10.1038/s41477-023-01476-z) suggested that there are “… two leading hypotheses: a single domestication event in China or multiple domestication events in geographically separate areas. These two hypotheses differ in their predicted history of genes/alleles selected during domestication..” “Analysis of dataset of 1,578 resequenced genomes and 993 selected genes that generated phylogenetic trees on which japonica and indica formed a monophyletic group, suggesting that the domestication alleles of these genes originated only once in either japonica or indica.”

“Importantly, the domestication alleles of most selected genes (~80%) stemmed from wild rice in China, but the domestication alleles of a substantial minority of selected genes (~20%) originated from wild rice in South and Southeast Asia, demonstrating separate domestication events of Asian rice”.

In comparison to these and other research works that employed vast amount of genetic and phenetic data, this research reports besides the morphological characteristics, only RAPD band patterns to assay the genetic polymorphism of rice grains found in Cheongju Sorori and compare it with wild rice, weed rice, and modern rice. Based on these results, the Authors have suggested the origin of rice domestication in Korea.

-- Thank you for your comments. The paper by Jing et al. (2023) entitled “Multiple domestications of Asian rice” was about two leading hypotheses: a single domestication event in China or multiple domestication events in geographically separate areas based on the analysis of dataset of the current rice germplasms (1,578 resequenced genomes) and 993 selected genes that generated phylogenetic trees on which japonica and indica. Our review paper is summarizing based on the investigation about 15,000 years old carbonized rice hulls excavated from the carbonized soil peat layers. I think it is hard to make a connection to the report with the analysis of the modern rice varieties, because there are too much time gaps and different environments between the current rice varieties and the 15,000 years old rice hulls. We really wanted to provide more detailed information on the carbonized ancient rice for this paper, but we could not carry out more experiments due to the limited experimental materials. I hope reviewer can understand the very weak situation with this old carbonized rice hulls which already destroyed by long years.

Reviewer cannot accept the reported here RAPD results as reliable for several reasons:

1)         DNA isolated from the ancient carbonized rice samples has very high risk (close to 100%) of contamination by microbial and fungal DNA of both old and more recent age. DNA sequencing must be applied to confirm the identity of these DNA samples.

2)         The resulting RAPD profiles (Fig. 11) yielded very few diverse bands for the ancient samples (31 specific bands in total, Table 4). In opposite, Figure 14 show nearly identical RAPD (7 primers) profiles for ancient samples of diverse origin – lines 11: Gawaji excavated rice, 12: Ancient short grain, 13: Quasi rice 1, 14: Quasi rice 1, 15: Ancient short grain, and 16: Quasi rice 2. At the same time, modern rice samples were variable. This fact has no any reasonable explanation.

-- Thank you for your comments. According to the reviewers comments, we decided to delete the PCR results derived from RAPD primers on pages 12~15, “2.3.5. DNA variation of the carbonized rice, Experiment 1:”.

3)         Reliability of the phylogenetic constructions shown in Figure 11, 15-17 cannot be evaluated because they are shown without any bootstrap of another statistical analysis. 

-- Thank you for your comments. We reanalyzed the results derived from the Universal Rice Primers (URP primers) on page 15~17 using the UPGMA method of POPTREE2 software (Takezaki 2009) as shown in Figure 13. And then we rewrote the section based on new analysis in pages 13~14 and 23. And Figure 16 was also deleted.

The manuscript describes a lot of data, but, the particular RAPD results cannot be accepted for publication.

-- Thank you for your comments. According to reviewers comments, we decided to delete the PCR results derived from RAPD primers.

It is highly recommended to remove this part from the manuscript.

-- Thank you for your comments. According to reviewers comments, we decided to delete the PCR results derived from RAPD primers. And Figure 16 was also deleted.

Round 3

Reviewer 1 Report

Comments and Suggestions for Authors

I thank the Authors for taking into consideration my previous observations upon the ms., which is now not encumbered by “genetic” speculations and is more to the point, representing, as I already said, a nice piece of work. A couple of final comments to tie it up:

It is not needed to state that “The URP primers are not RAPD (Randomly Amplied Polymorphic DNA) primers (line 365)”, please remove this sentence.

 Also, the legend to Figure 12 should state the distance method used: please substitute “The phylogenetic tree was constructed using the UPGMA method by POPTREE2 software.” with “The UPGMA tree displayed was built by using Nei’s standard genetic distance method.”

Comments on the Quality of English Language

It is almost OK.

Author Response

Response to Reviewer 1 _ R3

I thank the Authors for taking into consideration my previous observations upon the ms., which is now not encumbered by “genetic” speculations and is more to the point, representing, as I already said, a nice piece of work. A couple of final comments to tie it up:

It is not needed to state that “The URP primers are not RAPD (Randomly Amplied Polymorphic DNA) primers (line 365)”, please remove this sentence.

-- We appreciate reviewers kind comments. Yes, we deleted it.

 Also, the legend to Figure 12 should state the distance method used: please substitute “The phylogenetic tree was constructed using the UPGMA method by POPTREE2 software.” with “The UPGMA tree displayed was built by using Nei’s standard genetic distance method.”

-- We appreciate reviewers kind comments. Yes, we modified it as follows:

Figure 12. Phylogenetic analysis of ancient and current rice DNAs extracted from a single hull and amplified by universal rice primers, URP1, URP12, and URP13. The phylogenetic tree was constructed using the UPGMA method by POPTREE2 software with the UPGMA tree displayed by building using Nei’s standard genetic distance method.

Reviewer 3 Report

Comments and Suggestions for Authors

The manuscript titled "The Oldest ‘Cheongju Sorori Rice’ Excavated in Korea"  is devoted to justification of the theory that rice grains excavated from the Soori site are presumed to be the origin of rice domestication in Korea.

The manuscript is well-written and can be an excellent example of polyphasic research in plant domestication. But, one part, devoted to   DNA variation of the carbonized rice (Line 359) is still below critics.

Authors wrote: “In order to compare the genetic similarity with current rice by analyzing the DNA of  six types of carbonized rice seeds excavated from the peat layer of the Sorori site in  Cheongju, URP (universal rice primers) were used. The URP primers are not RAPD (Randomly Amplied Polymorphic DNA) primers, but the URP primers were derived from the rice repetitive DNA sequence of pKRD (GenBank accession No. AF241234). “

Note: All methods based on repetitive DNA sequences have a common methodological problem - they amplify anonymous DNA fragments of similar length, which may not have any DNA similarity at all.

This argument, along with empirical observations, suggests that PCR with arbitrary primers has limited application in molecular systematics beyond the intraspecific level. (Black, W.C., IV (1993), PCR with arbitrary primers: approach with care. Insect Molecular Biology, 2: 1-6. https://doi.org/10.1111/j.1365-2583.1993.tb00118.x). Moreover, this method is very sensitive to DNA degradation.

The main advantage of the polymerase chain reaction is its ability to detect and amplify minimal amounts of target DNA. However, in the context of DNA analysis, this advantage also represents a major danger regarding its reliability. This is because of the possible presence of exogenous DNA in ancient samples, which can cause false-positive results. As previously reported, most recoverable DNA from ancient specimens is bacterial or fungal in origin. (Francalacci, P. (1995). DNA recovery from ancient tissues: problems and perspectives. Human Evolution, 10(1), 81–91. doi:10.1007/bf02437517).

According to these experiments, DNA degradation occurs very soon after seed death, with 100-year-old seeds already containing highly degraded ribosomal RNA (rRNA) and DNA. Low molecular weight nucleic acids have been found in neolithic grains from Egyptian tombs (Cheah and Osborne, 1978). Rogers and Bendich (1985) extracted polymerized DNA from mummified seeds and seed embryos that were between 500 and 44,000 years old. However, they did not provide evidence for an endogenous origin of this DNA (Cited after FRANCO ROLLO, FRANCO MARIA VENANZI, and AUGUSTO AMICI: Dried Samples: Hard Tissues 16 DNA and RNA from Ancient Plant Seeds. In: B. Herrmann et al. (eds.), Ancient DNA © Springer-Verlag New York Inc. 1994.

In the work of Özgen M, Özdilek A, Birsin MA, et al. Analysis of ancient DNA from in vitro grown tissues of 1600-year-old seeds revealed the species as Anagyris foetida. Seed Science Research. 2012;22(4):279-286. doi:10.1017/S0960258512000207, and in work of Smirnova, N., Shipilina, L., and Khlestkina, E. 2023. Using DNA markers to reconstruct the lifetime morphology of barley grains from carbonized cereal crop remains unearthed at Usvyaty Settlement. Bio. Comm. 68(1): 3–9. https://doi.org/10.21638/spbu03.2023.101 and many others, Authors successfully used PCR to amplify the fragments of single genes of taxonomic importance from ancient samples.

The results obtained in the reviewed manuscript by URP primers, and presented at the Fig. 11 -12,  show low statistical significance of the similarity between the amplified samples (bootstrap values from 22 to 75%). 

It is recommended to remove this part from the manuscript. The text is enough large and significant, and it will benefit from removal of its weakest part.  

Author Response

Response to Reviewer 3 _ R3

The manuscript titled "The Oldest ‘Cheongju Sorori Rice’ Excavated in Korea"  is devoted to justification of the theory that rice grains excavated from the Soori site are presumed to be the origin of rice domestication in Korea.

The manuscript is well-written and can be an excellent example of polyphasic research in plant domestication. But, one part, devoted to DNA variation of the carbonized rice (Line 359) is still below critics.

Authors wrote: “In order to compare the genetic similarity with current rice by analyzing the DNA of six types of carbonized rice seeds excavated from the peat layer of the Sorori site in Cheongju, URP (universal rice primers) were used. The URP primers are not RAPD (Randomly Amplied Polymorphic DNA) primers, but the URP primers were derived from the rice repetitive DNA sequence of pKRD (GenBank accession No. AF241234). “

Note: All methods based on repetitive DNA sequences have a common methodological problem - they amplify anonymous DNA fragments of similar length, which may not have any DNA similarity at all.

This argument, along with empirical observations, suggests that PCR with arbitrary primers has limited application in molecular systematics beyond the intraspecific level. (Black, W.C., IV (1993), PCR with arbitrary primers: approach with care. Insect Molecular Biology, 2: 1-6. https://doi.org/10.1111/j.1365-2583.1993.tb00118.x). Moreover, this method is very sensitive to DNA degradation.

-- We appreciate the reviewer’s critical comments. The reviewer mentioned that repetitive DNA sequences have a common methodological problem: they amplify anonymous DNA fragments of similar length. However, the URP primers revealed multiple bands that allow DNA variation analysis among many varieties. They are commonly used for variety identification or grouping in various species, as reported by Kang et al. (Mol. Cells, 13(2), 2001; Mol Breeding 21, 2008). These studies described the usefulness of 20 primers with 20 nucleotides, referred to as universal rice primers (URP). Specifically, the papers titled “Fingerprinting of Diverse Genomes Using PCR with Universal Rice Primers Generated from Repetitive Sequence of Korean Weedy Rice” and “Genomic Characterization of Oryza Species-specific CACTA-like Transposon Element and Its Application for Genomic Fingerprinting of Rice Varieties” highlight their effectiveness.

We believe that these URP primers are sufficient to analyze the genetic variation in rice species for at least high-level clustering purposes.

URP primers are not arbitrary primers like RAPD.

* References

(1) Mol. Cells, 13(2): 281-287. (2001) Fingerprinting of Diverse Genomes Using PCR with Universal Rice Primers Generated from Repetitive Sequence of Korean Weedy Rice.

(2) Mol Breeding (2008) 21:283–292. Genomic characterization of Oryza species-specific CACTA-like transposon element and its application for genomic fingerprinting of rice varieties.

The main advantage of the polymerase chain reaction is its ability to detect and amplify minimal amounts of target DNA. However, in the context of DNA analysis, this advantage also represents a major danger regarding its reliability. This is because of the possible presence of exogenous DNA in ancient samples, which can cause false-positive results. As previously reported, most recoverable DNA from ancient specimens is bacterial or fungal in origin. (Francalacci, P. (1995). DNA recovery from ancient tissues: problems and perspectives. Human Evolution, 10(1), 81–91. doi:10.1007/bf02437517).

According to these experiments, DNA degradation occurs very soon after seed death, with 100-year-old seeds already containing highly degraded ribosomal RNA (rRNA) and DNA. Low molecular weight nucleic acids have been found in neolithic grains from Egyptian tombs (Cheah and Osborne, 1978). Rogers and Bendich (1985) extracted polymerized DNA from mummified seeds and seed embryos that were between 500 and 44,000 years old. However, they did not provide evidence for an endogenous origin of this DNA (Cited after FRANCO ROLLO, FRANCO MARIA VENANZI, and AUGUSTO AMICI: Dried Samples: Hard Tissues 16 DNA and RNA from Ancient Plant Seeds. In: B. Herrmann et al. (eds.), Ancient DNA © Springer-Verlag New York Inc. 1994.

-- We appreciate the reviewer’s critical comments. However, we would like to clarify that we used leaf DNA derived from current rice varieties, wild relatives, and weedy types of rice as controls. Although the DNA was extracted from ancient hulls, the DNA bands of the ancient hulls amplified with URP primers showed similar patterns to those of the controls, suggesting that there was no DNA contamination. If there had been contamination from non-rice sources, the DNA band patterns would have differed significantly from those of the current rice samples. Furthermore, in a recent experiment where we extracted DNA from rice hulls incubated in soil for two weeks, we did not observe any amplification of microorganism bands from the rice DNA samples.

Regarding DNA degradation, we understand that long periods can lead to DNA degradation in cells. However, rice hulls contain silica in their cell walls, making the DNA relatively more stable compared to human DNA, which is protected only by cell membranes in tombs.

In the work of Özgen M, Özdilek A, Birsin MA, et al. Analysis of ancient DNA from in vitro grown tissues of 1600-year-old seeds revealed the species as Anagyris foetida. Seed Science Research. 2012;22(4):279-286. doi:10.1017/S0960258512000207, and in work of Smirnova, N., Shipilina, L., and Khlestkina, E. 2023. Using DNA markers to reconstruct the lifetime morphology of barley grains from carbonized cereal crop remains unearthed at Usvyaty Settlement. Bio. Comm. 68(1): 3–9. https://doi.org/10.21638/spbu03.2023.101 and many others, Authors successfully used PCR to amplify the fragments of single genes of taxonomic importance from ancient samples.

The results obtained in the reviewed manuscript by URP primers, and presented at the Fig. 11 -12,  show low statistical significance of the similarity between the amplified samples (bootstrap values from 22 to 75%). 

It is recommended to remove this part from the manuscript. The text is enough large and significant, and it will benefit from removal of its weakest part.  

-- We greatly appreciate the reviewer’s insightful comments. Upon examining the gel images from the PCR amplification, we found that the band patterns are similar to those observed in current rice samples. If they had been derived from other species, such as microorganisms, the band patterns would have been markedly different from those of rice.

We would like to retain the URP results. To address the reviewer's concerns, we have removed the similarity values and retained only the grouping results to illustrate the clustering trends of the ancient rice hulls and Quasi rice. Accordingly, we have deleted all similarity values and revised the DNA results sections in “2.3.5. DNA Variation of the Carbonized Rice” and the “DNA Variation of Carbonized Rice” portion of the Discussion.

Reviewer 1 also agreed to retain the DNA variation parts with the URP analysis as follows:

*****************************************************************************************************

Response to Reviewer 1 _ R3

I thank the Authors for taking into consideration my previous observations upon the ms., which is now not encumbered by “genetic” speculations and is more to the point, representing, as I already said, a nice piece of work. A couple of final comments to tie it up:

It is not needed to state that “The URP primers are not RAPD (Randomly Amplied Polymorphic DNA) primers (line 365)”, please remove this sentence.

-- We appreciate reviewers kind comments. Yes, we deleted it.

 Also, the legend to Figure 12 should state the distance method used: please substitute “The phylogenetic tree was constructed using the UPGMA method by POPTREE2 software.” with “The UPGMA tree displayed was built by using Nei’s standard genetic distance method.”

-- We appreciate reviewers kind comments. Yes, we modified it as follows:

Figure 12. Phylogenetic analysis of ancient and current rice DNAs extracted from a single hull and amplified by universal rice primers, URP1, URP12, and URP13. The phylogenetic tree was constructed using the UPGMA method by POPTREE2 software with the UPGMA tree displayed by using Nei’s standard genetic distance method.

******************************************************************************************************
